# Deconstructing cold-induced brown adipocyte neogenesis in mice

Rayanne B Burl[1†], Elizabeth Ann Rondini[1,2†], Hongguang Wei[1,2], Roger Pique-Regi[1], James G Granneman[1,2]*

[1]Center for Molecular Medicine and Genetics, Wayne State University, Detroit, United States; [2]Center for Integrative Metabolic and Endocrine Research, Wayne State University, Detroit, United States

**Abstract** Cold exposure triggers neogenesis in classic interscapular brown adipose tissue (iBAT) that involves activation of β1-adrenergic receptors, proliferation of PDGFRA+ adipose tissue stromal cells (ASCs), and recruitment of immune cells whose phenotypes are presently unknown. Single-cell RNA-sequencing (scRNA-seq) in mice identified three ASC subpopulations that occupied distinct tissue locations. Of these, interstitial ASC1 were found to be direct precursors of new brown adipocytes (BAs). Surprisingly, knockout of β1-adrenergic receptors in ASCs did not prevent cold-induced neogenesis, whereas pharmacological activation of the β3-adrenergic receptor on BAs was sufficient, suggesting that signals derived from mature BAs indirectly trigger ASC proliferation and differentiation. In this regard, cold exposure induced the delayed appearance of multiple macrophage and dendritic cell populations whose recruitment strongly correlated with the onset and magnitude of neogenesis across diverse experimental conditions. High-resolution immunofluorescence and single-molecule fluorescence in situ hybridization demonstrated that cold-induced neogenesis involves dynamic interactions between ASC1 and recruited immune cells that occur on the micrometer scale in distinct tissue regions. Our results indicate that neogenesis is not a reflexive response of progenitors to β-adrenergic signaling, but rather is a complex adaptive response to elevated metabolic demand within brown adipocytes.

**\*For correspondence:**
jgranne@med.wayne.edu

[†]These authors contributed equally to this work

**Competing interest:** The authors declare that no competing interests exist.

## Editor's evaluation

This study elucidates transcriptional profiles of the stromal vascular fraction of murine brown adipose tissue in the context of thermogenic stimulation. The authors combine systems and reductionist approaches to show the reliance of mature brown adipocytes on adrenergic activation to indirectly stimulate progenitor proliferation and differentiation. This timely work will provide beneficial data for public use and further resolve the complexities underlying brown adipose physiology.

## Introduction

Brown adipose tissue (BAT) is a specialized organ that is the dominant site of non-shivering thermogenesis in neonatal mammals (*Dawkins and Hull, 1964*). While critical for regulation of neonatal body temperature, adult mammals, including humans, also have metabolically active BAT (*Cypess et al., 2009*; *Lee et al., 2013b*; *Sacks and Symonds, 2013*). Importantly, adult BAT mass and function are associated with a healthier metabolic profile in both rodents and humans (*Becher et al., 2021*; *Wibmer et al., 2021*; *Herz et al., 2021*). Although the direct mechanisms have not been fully elucidated, BAT abundance and activity is associated with higher energy expenditure, altered secretion of hormones (*Chondronikola et al., 2016*; *Géloën et al., 1988*; *Hanssen et al., 2015*; *Lee et al., 2015*; *Villarroya et al., 2017*; *Villarroya et al., 2019*), and reduced visceral fat mass (*Herz et al., 2021*).

Therefore, a more thorough understanding of the processes that regulate the activation and physiological expansion of BAT could have important therapeutic implications for obesity-related metabolic disease.

Cold exposure triggers BAT neogenesis and is an important adaptive means to increase the capacity for non-shivering thermogenesis in rodents (*Bukowiecki et al., 1982*; *Bukowiecki et al., 1986*; *Foster and Frydman, 1978*; *Nedergaard, 1982*; *Nedergaard et al., 2019*). Fate mapping studies in rats by Bukowiecki et al. strongly indicated that brown adipocytes (BAs) arise from interstitial cells that proliferate and differentiate during the first few days of cold exposure (*Bukowiecki et al., 1986*; *Géloën et al., 1988*). More recently, genetic lineage-tracing experiments from our lab demonstrated that most, if not all, BAs induced by cold arise from stromal cells that express the surface marker platelet-derived growth factor receptor alpha (PDGFRA). In addition to effects on brown adipogenesis, cold exposure also remodels the vasculature, recruits monocytes/macrophages, and triggers proliferation of uncharacterized population(s) of cells that are potentially important for proper BAT expansion and remodeling (*Lee et al., 2015*; *Géloën et al., 1988*).

The complete cellular complexity of BAT is presently unknown. In this regard, recent single-cell transcriptional analysis indicates the existence of multiple stromal cell types in white adipose tissue (WAT) depots that express PDGFRA (*Burl et al., 2018*; *Hepler et al., 2018*; *Merrick et al., 2019*; *Schwalie et al., 2018*; *Emont et al., 2022*; *Rondini and Granneman, 2020*; *Sárvári et al., 2021*; *Rondini et al., 2021*), and that recruitment of brown/beige adipocytes in response to β3-adrenergic activation involves specific subsets of stromal cells and immune cells. In addition, we previously observed that cold triggers proliferation of cells expressing the myeloid cell surface marker F4/80, as well as uncharacterized population(s) of stromal cells (*Lee et al., 2015*). The simultaneous proliferation of multiple cell types within BAT suggests that cold-induced neogenesis involves the functional interaction of numerous cell types, as has been demonstrated for BA neogenesis in WAT (*Lee and Granneman, 2012*; *Lee et al., 2016*; *Lee et al., 2013a*; *Lee et al., 2014*). Curiously, cold-induced mitogenesis in rodent BAT only commences after more than 2 days of cold exposure (*Bukowiecki et al., 1982*; *Bukowiecki et al., 1986*; *Hunt and Hunt, 1967*; *Lee et al., 2015*) and is concentrated within specific regions of the tissue (*Lee et al., 2015*). The basis of the timing and location of cold-induced brown adipogenesis is not known, nor is the relationship, if any, to proliferating immune cells.

Cold-induced neogenesis in BAT requires intact sympathetic activity that can be mimicked by systemic norepinephrine (NE) infusion in warm-adapted mice (*Lee et al., 2015*). Moreover, the effects of NE infusion are prevented by global knockout (KO) of the β1-adrenergic receptor (ADRB1) (*Lee et al., 2015*). ADRB1, but not β3-adrenergic receptors (ADRB3), are present on ASC, and in vitro experiments suggested that activation of preadipocyte ADRB1 mediates cold-induced BA neogenesis (*Bronnikov et al., 1999*; *Bronnikov et al., 1992*). However, ADRB1 is expressed on additional cell types, including mature BAs. Therefore, whether the effects of cold on ASC proliferation are directly mediated by ADRB1 signaling in ASCs or indirectly by metabolic activation of BAs remains unclear.

To address these unresolved questions, we profiled global gene transcription in BAT to identify the duration of cold exposure that captures peak cellular proliferation and differentiation. We observed that the timing and magnitude of cold-induced neogenesis varied among individual mice, but was nonetheless strongly predicted by immune cell recruitment, independent of time in the cold. We then performed comprehensive single-cell RNA-sequencing (scRNA-seq) of mouse interscapular BAT (iBAT) stromal cells under control conditions, and at the peak of cold-induced proliferation and differentiation. scRNA-seq identified three major *Pdgfra*-expressing subtypes, one of which, termed adipose tissue stromal cell 1 (ASC1), was highly responsive to cold exposure and appears to exclusively contribute to preadipocyte proliferation and BA differentiation in vivo. scRNA-seq also revealed that cold exposure recruits several immune cell types to iBAT, including specialized lipid-handling macrophages and diverse populations of dendritic cells. Immunofluorescence and single-molecule fluorescence in situ hybridization (smFISH) analysis demonstrated that cold exposure recruits immune cells to distinct tissue locations near the periphery of the iBAT and sites within the central parenchyma where BAs undergo efferocytosis. Importantly, we demonstrate by high resolution confocal imaging of genetic and chemical tracers that cold-induced adipogenic niches are created and resolved by the dynamic interactions of ASC1 and recruited immune cells.

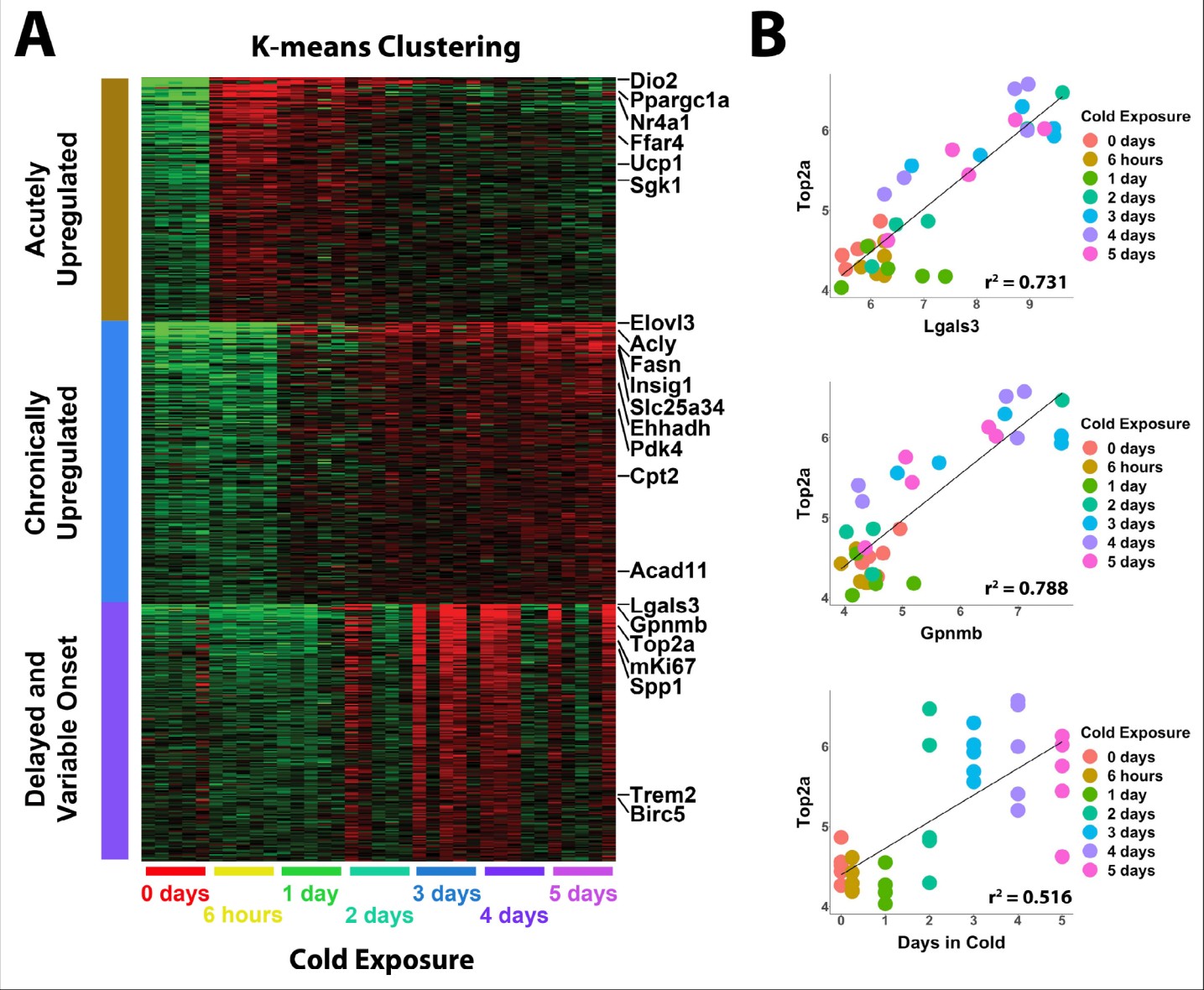

**Figure 1.** Whole tissue RNA-sequencing analysis reveals proliferation correlates with immune cell recruitment during cold exposure. (**A**) Heatmap of K-means clustering of whole tissue RNA-sequencing data. Rows of the heatmap are genes, and columns are individual RNA-seq libraries. Red and green colors represent upregulation and downregulation, respectively. Analysis includes five replicates (individual mice) each from seven different cold exposure durations: room temperature controls, and 6 hr, 1 day, 2 days, 3 days, 4 days, or 5 days of cold exposure for a total of 35 RNA-seq libraries. (**B**) Correlation of specific variables in the RNA-seq data with *Top2a* expression by individual library. r² values are displayed on the plot.

## Results

### RNA-seq analysis of mouse iBAT during a cold exposure time course reveals activation of immune cells that correlates with proliferation

To gain insight into the timing of cold-induced iBAT neogenesis, we first sequenced total tissue RNA to establish the time course and profile individual variation of cold-induced gene expression (*Figure 1*). K-means clustering of the top differentially expressed genes (DEGs) identified three distinct patterns of upregulated gene expression (*Figure 1A*). Acutely upregulated genes were induced within 6 hours of cold exposure, but then returned to control levels after chronic cold acclimation. Genes in this cluster are well-known targets of protein kinase A (PKA), including peroxisome proliferative activated receptor gamma coactivator 1 alpha (*Ppargc1a*), uncoupling protein 1 (*Ucp1*), nuclear receptor subfamily 4 group A member 1 (*Nr4a1*), and iodothyronine deiodinase 2 (*Dio2*) (*Figure 1A*). Chronically

upregulated genes were induced following 1–2 days of cold exposure and remained elevated for the duration of the time in cold. This cluster included genes involved in lipid synthesis and oxidation, such as elongation of very long chain fatty acids-like 3 (*Elovl3*), fatty acid synthase (*Fasn*), and pyruvate dehydrogenase kinase isoenzyme 4 (*Pdk4*). The third cluster was interesting, as the gene upregulation was delayed and variable among individual mice. Genes in this cluster included proliferation markers (*Birc5*, *Top2a*), as well as various genes indicating innate immune activation and macrophage recruitment (*Lgals3*, *Gpnmb*, *Trem2*) (*Figure 1A*). Importantly, these variables were highly correlated and this relationship was largely independent of time spent in the cold (*Figure 1B*; p<0.001). These data confirm that cell proliferation and immune cell recruitment are connected and peak around the fourth day of cold exposure.

## iBAT scRNA-seq of total stromal cells identifies multiple stromal cell subtypes

To investigate heterogeneity of iBAT stromal and immune cells, as well as gain insight into adipogenic differentiation in vivo, we performed scRNA-seq analysis of stromal cells isolated from iBAT (*Figure 2—figure supplement 1A*). Mice were adapted to room temperature (RT; 22–23 °C) or exposed to 6 °C for 4 days to induce iBAT neogenesis and capture the peak in cold-induced proliferation. iBAT stromal cells were isolated and cells were split into immune and non-immune cell populations by magnetic bead cell separation (MACS) with a lineage marker cocktail (*Figure 2—figure supplement 1A*). Individual single-cell libraries were prepared from two independent experiments of RT control and cold-exposed mice, yielding a total of eight single-cell libraries (*Figure 2—figure supplement 1C*). Sequencing data from these independent cohorts were merged and integrated, as detailed in Materials and methods (*Figure 2—figure supplement 1B*).

Lineage marker negative (Lin-) cell libraries contained adipose stromal cells (*Pdgfra*+ ASCs), vascular cells, and proliferating/newly differentiating adipocytes (*Figure 2A*). Clustering of scRNA-seq data from control and cold-exposed mice identified eight major clusters, ranging from ~500 to 6,500 cells per cluster (*Figure 2A*, *Supplementary file 1*). Three of these clusters were defined as ASCs based on their expression of common mesenchymal stem cell markers *Pdgfra*, *Cd34*, and lymphocyte antigen 6 complex locus A (*Ly6a*, a.k.a. Sca1) (*Figure 2—figure supplement 2A*) that are often used for identification of adipocyte progenitors (*Burl et al., 2018*; *Hepler et al., 2018*; *Merrick et al., 2019*; *Schwalie et al., 2018*). ASCs clustered together at low resolution (resolution < or = 0.04), indicating they are more similar to each other than the other cell types present in the libraries (*Figure 2—figure supplement 2B*). scRNA-seq also identified a cluster of proliferating and differentiating ASCs (Prolif/Diff) (*Figure 2A*). The remaining clusters appeared to be a mixture of vascular endothelial cells (VEC), vascular smooth muscle cells (VSMC), Schwann cells, and a small (~5%) mixture of immune cells that were not excluded by MACs separation (*Figure 2A*). Separating data by treatment revealed that two cell clusters were unique to cold-exposed mice (circled; *Figure 2B*). One cluster retained numerous ASC markers like *Pdgfra*, whereas the other was largely *Pdgfra* negative and expressed high levels of markers of proliferation (e.g. *Birc5*) and adipocyte differentiation (e.g. *Car3*) (*Figure 2—figure supplement 2A*, *Supplementary file 1*).

To gain greater insight into the relationships among the various ASCs, we reclustered the ASC and Prolif/Diff populations at a higher resolution (*Figure 2C*, *Supplementary file 2*). This clustering resolved three distinct ASC cell types prominent in iBAT controls: ASC1-3 (*Figure 2C*, left). Genes that define these clusters were similar to the expression profiles of mouse PDGFRA+ ASC subtypes recently identified in various mouse fat depots (*Burl et al., 2018*; *Dong et al., 2022*; *Hepler et al., 2018*; *Merrick et al., 2019*; *Rondini and Granneman, 2020*; *Schwalie et al., 2018*). In control mice, ASC subtypes were distinguished by genes that encode extracellular matrix (ECM) and matrix remodeling proteins, and paracrine signaling proteins of the transforming growth factor beta superfamily. Thus, ASC1 selectively expressed collagen type V alpha 3 chain (*Col5a3*), C-X-C motif chemokine ligand 14 (*Cxcl14*), and the bone morphogenic protein (BMP)-binding endothelial regulator (*Bmper*) (*Figure 2D*). Cells in ASC2 expressed secreted protease inhibitor peptidase inhibitor 16 (*Pi16*), surface glycoprotein dipeptidyl peptidase 4 (*Dpp4*), and the ECM component fibronectin (*Fbn1*). ASC3 cells selectively expressed secreted ligand growth differentiation factor 10 (*Gdf10*), C-type lectin domain containing 11a (*Clec11a*), and fibulin1 (*Fbln1*) (*Figure 2D*).

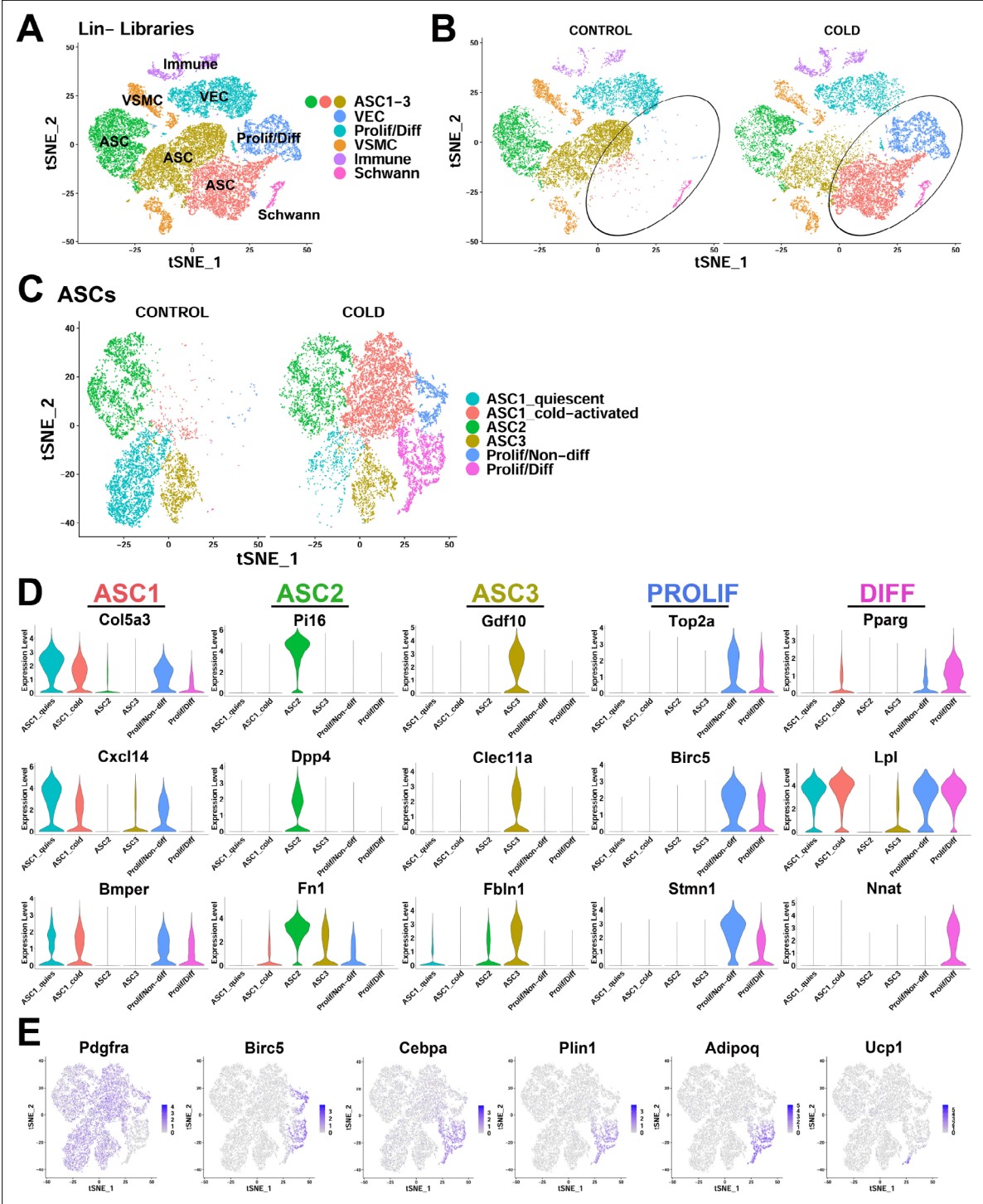

**Figure 2.** scRNA-seq reveals ASC heterogeneity and maps adipogenic trajectories in mouse iBAT. (**A**) t-SNE plot of 28,691 lineage marker negative (Lin-) cells from iBAT of control mice and mice exposed to cold for four days. Clustering identified eight major clusters, highlighted in different colors. ASC, adipose tissue stromal cell; VEC, vascular endothelial cell; VSMC, vascular smooth muscle cells; Prolif/Diff, proliferating/differentiating cells. DEGs that define these clusters are in ***Supplementary file 1***. (**B**) t-SNE plot from (**A**) split into cells from the separate treatments (CONTROL and COLD). Circles highlight cold-induced clusters. (**C**) t-SNE plot of 19,659 re-clustered ASC and Prolif/Diff cells from (**A**). The t-SNE plot and clustering identified six clusters. Prolif/Non-diff, proliferating/non-differentiating; Prolif/Diff, proliferating/differentiating. DEGs that define these clusters are in ***Supplementary file 2***. (**D**) Violin plots of log2 expression levels of select marker genes for individual clusters from the CONTROL and COLD data presented in (**C**). (**E**) t-SNE plots displaying the log2 expression levels for genes involved in adipogenic differentiation from the CONTROL and COLD data presented in (**C**).

The online version of this article includes the following figure supplement(s) for figure 2:

*Figure 2 continued on next page*

*Figure 2 continued*

**Figure supplement 1.** Schematic diagram of single-cell library generation and processing.

**Figure supplement 2.** scRNA-seq analysis of Lin- cells from iBAT of control and cold-exposed mice.

High-resolution clustering identified three additional cell clusters in cold-exposed mice (*Figure 2C*, right). All of these clusters expressed ASC1-specific markers; however, two of the clusters were primarily defined by genes for proliferation (*Top2a*, *Birc5*, *Stmn1*), and/or adipogenic differentiation (*Pparg*, *Lpl*, *Nnat*) (*Figure 2D*). This dramatic change in the expression profile of cells expressing ASC1 markers (e.g. *Col5a3*, *Cxcl14*, and *Bmper*) (*Figure 2—figure supplement 2C*) allowed for the characterization of these clusters as distinct expression states of the ASC1 subtype. For the purpose of exposition, we referred to these clusters as 'quiescent' and 'cold-activated' ASC1 from control and cold-exposed libraries, respectively (*Figure 2C*). Cold exposure had comparatively little impact on the profiles of ASC2 or ASC3 (*Figure 2—figure supplement 2C*). Examination of the two ASC1 expression states indicated that cold activation greatly reduced expression of genes involved in cholesterol biosynthesis (sterol biosynthetic process; GO:0016126, p=2.8E-10), cell adhesion (GO:0007155, p=4.9E-2) and extracellular matrix organization (GO:0030198, p=7.7E-7), and strongly induced expression of genes involved in immune system process (GO:0002376, p=2.2E-6), chemokine activity (GO:0008009, p=5.0E-3) and cell migration (GO:0016477, p=3.6E-4). In addition, ASC1 cells appeared highly poised for adipogenesis, expressing higher levels of the master adipocyte transcriptional regulator peroxisome proliferator activated receptor gamma (*Pparg*) and its target genes, such as lipoprotein lipase (*Lpl*) (*Figure 2D*). Notably, cells in the differentiating cluster selectively expressed the imprinted gene neuronatin (*Nnat*) that was transiently upregulated during differentiation and silenced in mature BAs (*Figure 2D*).

## iBAT scRNA-seq identifies cells undergoing cold-induced adipogenic differentiation

Numerous ASC1-specific marker genes were co-expressed in the proliferating and differentiating clusters (*Figure 2D*), indicating that expression of these genes persists as cell differentiated into BAs. In contrast, none of the aforementioned ASC2 and ASC3 markers were expressed in these proliferating/differentiating clusters (*Figure 2D*). From these data, we concluded that cold-activated ASC1 are the immediate progenitors of new BAs induced by cold exposure. In addition, differentiating cells exhibited a clear trajectory along t-SNE2 that included loss of ASC marker expression (*Pdgfra*), transient proliferation (*Birc5*), and sequential upregulation of early (*Cebpa*) and late (*Adipoq*, *Ucp1*) markers of brown adipogenesis (*Figure 2E*). In contrast, the proliferating, non-differentiating cells that retained ASC1 marker expression did not appear to contribute to adipogenesis and might function to replenish the ASC1 population, as suggested by previous fate mapping studies (*Lee et al., 2015*).

In summary, analysis of BAT ASCs and total stromal cell populations indicates that interstitial ASC1 cells are highly responsive to cold exposure and comprise most, if not all, BA progenitors during acute cold-induced neogenesis.

## Localizing ASC subtypes and adipogenic niches within the tissue microenvironment

scRNA-seq does not retain the tissue architecture and spatial relationships among cell subtypes, yet previous work suggested that cold-induced neogenesis occurs in specific tissue regions (*Lee et al., 2015*). To address this issue, we used scRNA-seq data to identify subtype-specific mRNAs for spatial analysis by smFISH. scRNA-seq data indicate that ASC1-3 are distinguished by the differential expression of ECM proteins and paracrine signaling factors, suggesting that these cells have distinct functions in the tissue microenvironment. Therefore, we examined the spatial distribution of ASCs by smFISH using the subtype-specific markers *Bmper* (ASC1), *Pi16* (ASC2), and *Gdf10* (ASC3) (*Figure 3A*). Using smFISH in combination with Pdgfra-CreER^T2 x LSL-tdTomato genetic tracing, we note that PDGFRA+ cells are found throughout the tissue, including the parenchyma and fascia. Using smFISH, we found that while ASC1 comprised the majority of the PDGFRA+ parenchymal interstitial cells, ASC2 were localized to the tissue fascia and surrounding large vessels, whereas ASC3 were predominately localized to areas surrounding vessels, but not capillaries (*Figure 3B–C*). Importantly, the interstitial

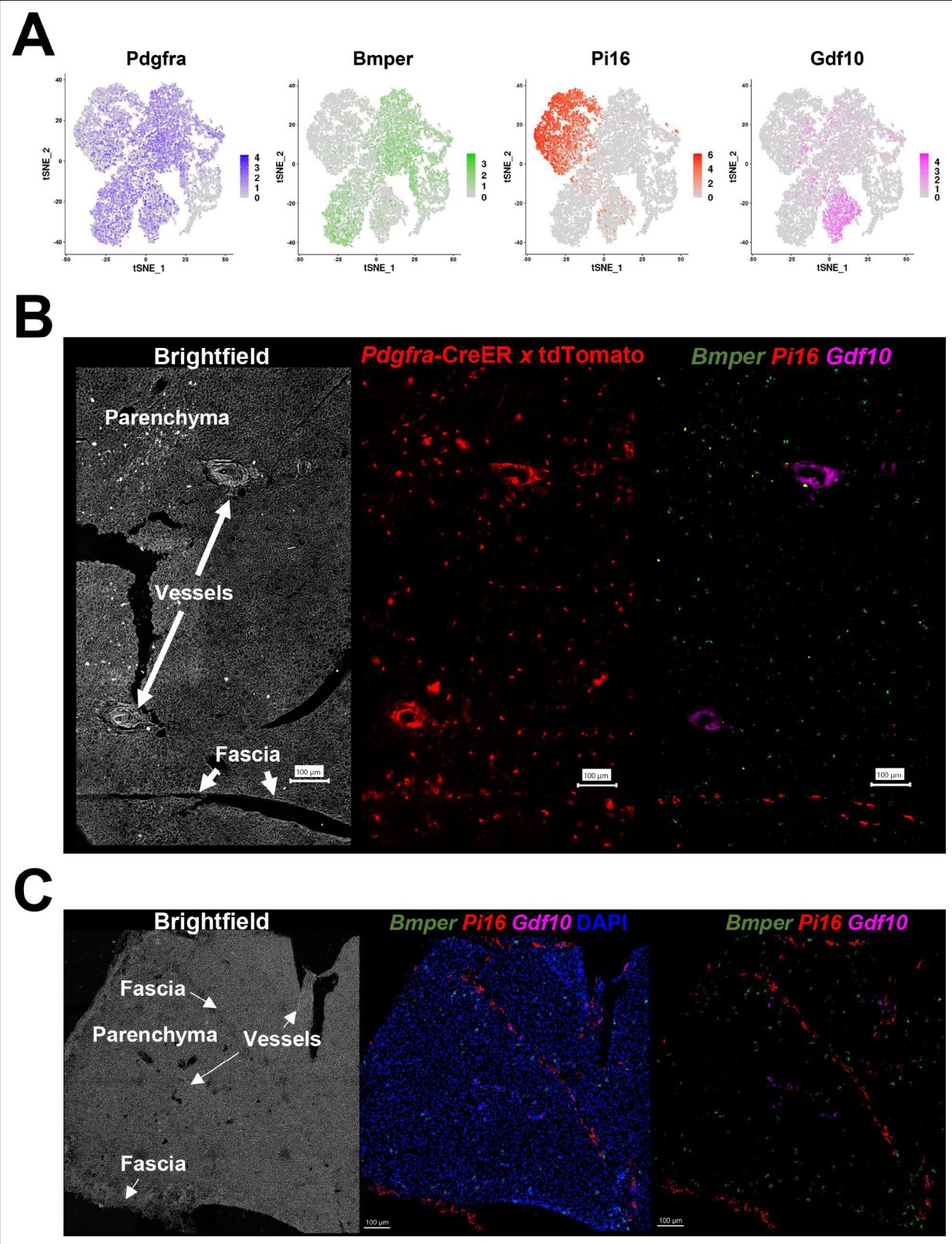

**Figure 3.** *Pdgfra*+ ASC subtypes occupy distinct areas of the tissue. (**A**) t-SNE plot of log2 gene expression from *Pdgfra* genetic tracing and smFISH probes in scRNA-seq data. t-SNE plot is ASCs from iBAT of control and cold-exposed mice, as in *Figure 2C*. (**B**) Representative image of fixed frozen iBAT from Pdgfra-CreER[T2] x LSL-tdTomato reporter mice. (Left) Brightfield image shows gross tissue structures, including the tissue fascia, parenchyma, and large vessels. (Center) TdTomato (red) native fluorescence. (Right) Tissue was bleached and stained with smFISH probes *Bmper* (green), *Pi16* (red),

*Figure 3 continued on next page*

Figure 3 continued

and *Gdf10* (pink). *Bmper* distinguishes ASC1, *Pi16* ASC2, and *Gdf10* ASC3. Scale bar, 100 µm. (**C**) Representative images of control fixed frozen iBAT stained with smFISH probes *Bmper* (green), *Pi16* (red), and *Gdf10* (pink) taken at higher resolution. Associated brightfield image shows gross tissue structures. Nuclei were counterstained with DAPI. Scale bar, 100 µm.

location of ASC1 is consistent with the work of *Bukowiecki et al., 1986* who, using electron microscopy, concluded that cold-induced brown adipocytes are derived from interstitial stromal cells.

## Mapping an adipogenic trajectory in situ

scRNA-seq data indicated an adipogenic trajectory in which ASC1 cells reduce expression of stromal markers (*Pdgfra* and *Dcn)*, while upregulating proliferation (*Top2a, Birc5*) and early differentiation markers (*Car3* and *Plin1*), and finally the terminal differentiation marker *Ucp1*. Among genes expressed within the adipogenic trajectory, we found that *Nnat* was induced during early differentiation, then silenced upon terminal differentiation (*Figure 4A*). Thus, *Nnat* expression marks the transient state of early differentiation. To determine whether we could observe an adipogenic trajectory in situ, we used multiplexed smFISH to probe for quiescent ASC1 (interstitial cells, *Dcn+ Top2a-*), proliferating ASC1 (interstitial cells, *Dcn+ Top2a+*), and early differentiating cells (*Nnat+ Top2a+/- Dcn* low) (*Figure 4B*). Low magnification imaging showed a fairly uniform distribution of *Dcn+* ASCs throughout control iBAT, with little evidence of proliferation (*Top2a*) or active differentiation (*Nnat*). In contrast, cold exposure triggered the appearance of numerous *Top2a+ Dcn+* cells, as well as clusters of *Nnat* positive cells with and without co-expression of *Top2a* (*Figure 4B*). Immunofluorescence analysis of NNAT protein confirmed NNAT+ cells containing nascent PLIN1+ lipid droplets (*Figure 4—figure supplement 1A*).

High-resolution three-dimensional confocal imaging provided strong evidence for an adipogenic trajectory within a tissue niche. Thus, we observed proliferating ASC1 immediately adjacent to proliferating/differentiating ASC1, and more distally to differentiating ASC1 lacking expression of proliferation markers. Note the concomitant loss of the ASC marker (*Dcn*) as cells undergo early differentiation (*Nnat*) (*Figure 4C*). As anticipated from scRNA-seq data, the majority of *Nnat+* cells co-expressed the ASC1 marker *Bmper*, but not the ASC2 marker *Pi16* (*Figure 4D* and *Figure 4—figure supplement 1B*).

## ASC proliferation/differentiation is triggered indirectly via adrenergic activation of BAs

Cold-induced neogenesis in BAT requires intact sympathetic innervation and can be mimicked by infusion of NE (*Géloën et al., 1992*; *Lee et al., 2015*). Furthermore, global knockout of ADRB1 blocked neogenesis induced by systemic NE infusion (*Lee et al., 2015*). In our iBAT single-cell data, *Adrb1* is expressed in ASC1, proliferating/differentiating ASCs, VSMCs, and sparsely in some immune cells (*Figure 5—figure supplement 1A*). Based on these observations and previous findings, we and others hypothesized that ADRB1 on preadipocytes mediates cold-induced proliferation. To test this hypothesis, we used Pdgfra-CreER^T2 to inducibly knockout *Adrb1* in PDGFRA+ cells from floxed Adrb1 (*Adrb1*^fl/fl^) mice and performed scRNA-seq analysis (*Figure 5A* and *Supplementary file 3*). Control mice were *Adrb1*^fl/fl^ mice treated with tamoxifen. Although sparsely expressed, scRNA-seq data indicates that *Adrb1* is expressed in ASC1, but not in ASC2 or ASC3 (*Figure 5B*). Tamoxifen treatment reduced *Adrb1* expression in ASC1 by more than 90% (chi-square p<0.00001, *Figure 5B*). Surprisingly, we found that knockdown of *Adrb1* in ASC1 had no discernable effect on the ability of cold exposure to increase ASC1 proliferation/differentiation (both being ~10 fold over basal in WT and *Adrb1* KO mice) (*Table 1*, *Supplementary file 4*, and *Figure 5C*).

In a larger independent cohort of mice, we found that *Adrb1* knockout in ASCs had no significant effect on cold-induced expression of proliferation makers at the whole tissue level (*Figure 5D*). In addition, smFISH demonstrated that cold induced the appearance of differentiating *Nnat+* preadipocytes in the absence of detectible *Adrb1* mRNA following Pdgfra-CreER^T2-mediated recombination (p<0.00001; *Figure 5—figure supplement 1B, C*). Taken together, these data demonstrate that ASC *Adrb1* expression is not required for cold-induced BA neogenesis. However, while *Adrb1* expression did not impact neogenesis, 68–72% of the total individual variation in proliferation marker expression (*Birc5, Top2a*) across treatment groups was accounted for by variation in markers of immune cell recruitment (*Trem2, Gpnmb*; *Figure 5E*).

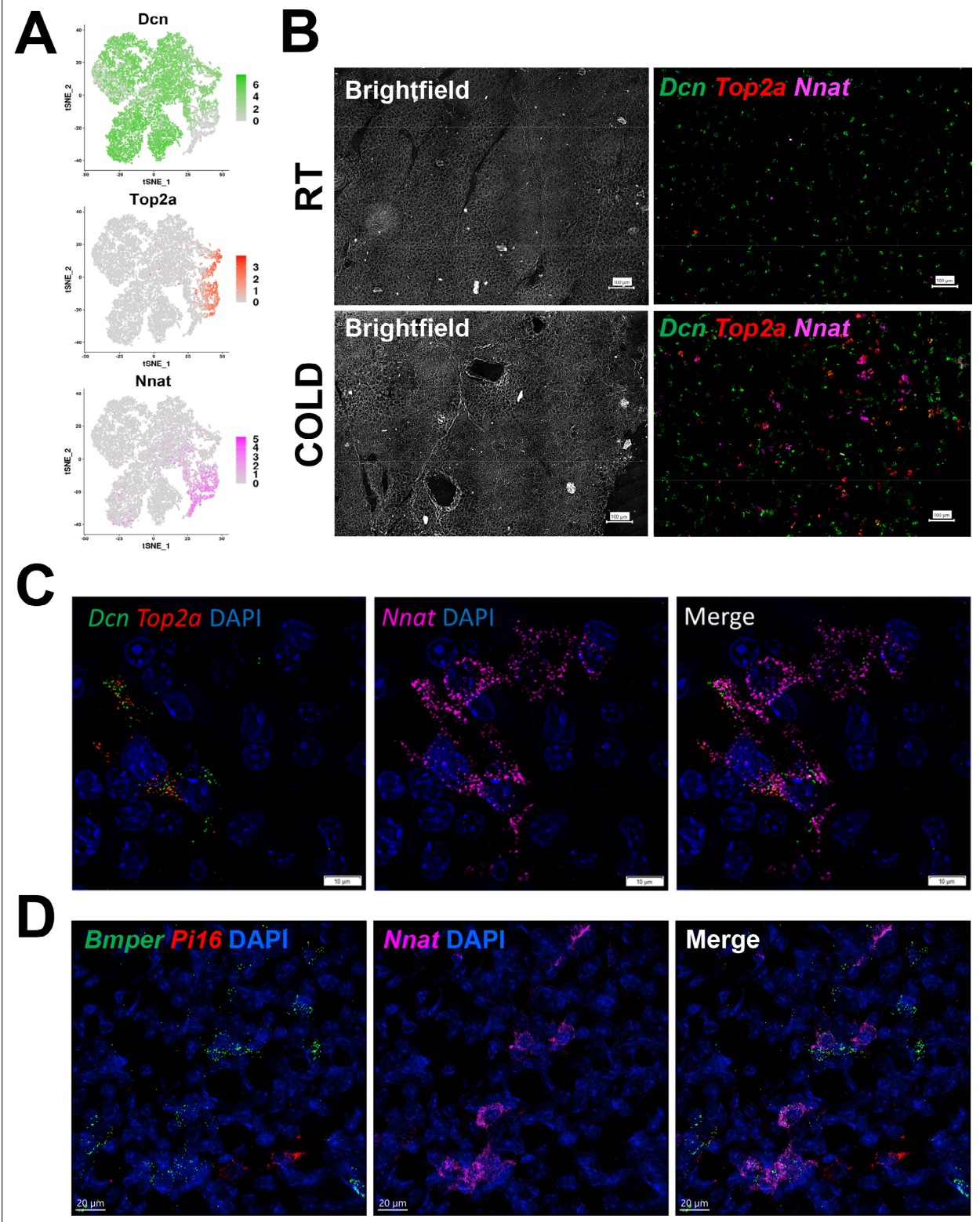

**Figure 4.** smFISH maps adipogenic trajectories and supports ASC1 as the direct precursors of new BA. (**A**) t-SNE plot of log2 gene expression of smFISH probes in scRNA-seq data. t-SNE plot displays ASCs from iBAT of control and cold-exposed mice. (**B**) Representative low magnification images of fixed frozen iBAT stained with smFISH probes for *Dcn* (green), *Top2a* (red), and *Nnat* (pink). Tissue is from control and cold-exposed mice, as indicated. Associated brightfield image shows gross tissue structure. Scale bar, 100 μm. (**C**) High-magnification image of fixed frozen cold-exposed mouse iBAT stained with smFISH probes *Dcn* (green), *Top2a* (red), and *Nnat* (pink). Nuclei were counterstained with DAPI. Scale bar, 10 μm. (**D**) High-

*Figure 4 continued on next page*

*Figure 4 continued*

magnification image of fixed frozen cold-exposed mouse iBAT stained with smFISH probes *Bmper* (green), *Pi16* (red), and *Nnat* (pink). Nuclei were counterstained with DAPI. Scale bar, 20 µm.

The online version of this article includes the following source data and figure supplement(s) for figure 4:

**Figure supplement 1.** Immunohistochemistry identified newly differentiating ASCs that are NNAT and PLIN1 positive.

**Figure supplement 1—source data 1.** Quantification of co-expression of the smFISH probe *Nnat* with probes for *Bmper* and *Pi16* (n=3 animals).

BAs also express ADRB1 and, owing to its higher affinity for NE, is a major receptor for mediating cold-induced metabolic activation (*Bukowiecki et al., 1978*; *Chaudhry and Granneman, 1999*). Additionally, we note that proliferation marker expression did not correspond to the rapid activation of classic adrenergic PKA targets, but instead required chronic cold exposure that was correlated with immune cell recruitment (*Figure 1A–B*). Thus, we reasoned that the effects of cold on ASC1 proliferation might be mediated indirectly by metabolic activation of BAs, similar to brown adipogenesis in WAT (*Lee et al., 2013a*; *Lee et al., 2012*). To determine whether adrenergic activation of BAs is sufficient to induce neogenesis, we infused 2 independent cohorts of mice with the highly-selective ADRB3 agonist CL316,243 (CL, 0.75 nmol/hr) for 4 days, noting that BAs express ADRB3, but ASCs do not (*Figure 5—figure supplement 1A*). We found that CL treatment triggered significant activation, proliferation and differentiation of ASC1 cells (*Table 1*, *Supplementary file 5*, *Supplementary file 6*, and *Figure 5—figure supplement 1D-F*). Taken together, these data strongly suggest that cold exposure triggers progenitor proliferation/differentiation *indirectly* via adrenergic activation of BAs and not through direct ADRB1 activation of progenitors.

## Cold exposure recruits macrophages and dendritic cells in iBAT

Previous analysis indicated that cold-induced neogenesis involves immune cell recruitment and proliferation of uncharacterized myeloid cells (*Lee et al., 2015*). To gain insight into the immune cell complexity of mouse iBAT, we used scRNA-seq to profile lineage positive (Lin+) cells from mice at RT or exposed to cold and identified 16 immune clusters (*Figure 6A* and *Supplementary file 7*). Seven of these clusters were different subpopulations of macrophages and dendritic cells (MAC and DEND, respectively), comprising 52.0% of the cells in this dataset. This dataset also contained populations of monocytes (MONO), natural killer T cells (NKT), T lymphocytes (Tlym), B lymphocytes (Blym), reticulocytes (RET), neutrophils (NEUT), and small populations of non-immune cells (<3% of total cells). The DEGs that define the Lin+ clusters are presented in *Supplementary file 7*.

Cold exposure dramatically altered the immune cell landscape in mouse iBAT and led to the expansion of dendritic cells and altered phenotype/subtypes of macrophages (*Figure 6B*, circled). All four dendritic cell clusters increased with cold exposure (DEND1-4; *Figure 6B*). DEGs for these clusters generally correspond to known dendritic cell subtypes. DEND1 cells expressed conventional dendritic cell type 1 (cDC1) markers, including high levels of the master transcriptional regulator *Irf8*, as well as genes *Ppt1*, *Jaml*, and *Naaa* (*Figure 6—figure supplement 1A*; *Guilliams and van de Laar, 2015*; *Hilligan and Ronchese, 2020*). These cDC1 cells were 33.6% of the dendritic cells. We identified the cluster DEND2 as monocyte-derived dendritic cells (moDCs) based on expression of the moDC markers *Cd209a* and *Itgam* (CD11b) (*Figure 6—figure supplement 1A*; *Cheong et al., 2010*). The most abundant dendritic cell type in this dataset (51.6% of dendritic cells), moDCs also expressed *Tnip3* and *Clec4b1* (*Figure 6—figure supplement 1A*). The minor dendritic cells clusters, DEND3 (12.1%) and DEND4 (2.7%) exhibited profiles consistent with mature/migratory dendritic cells (*Tmem123*, *Ccr7*) (*Förster et al., 2008*; *Takekoshi et al., 2010*), and plasmacytoid dendritic cells (*Irf8* high, *Cd7*, *Tcf4*) (*Rodrigues et al., 2018*; *Figure 6—figure supplement 1A*), respectively.

Previous fluorescence-activated cell sorting (FACS) analysis demonstrated that cold exposure induces proliferation in F4/80+ immune cells (*Lee et al., 2015*). While *Adgre1* (gene encoding the F4/80 antigen) was expressed in both macrophage and dendritic cell populations (*Figure 6—figure supplement 1B*), multidimensional scRNA-seq analysis indicates cold-induced proliferation occurs mostly in cDC1 (DEND1) and moDC (DEND2) subpopulations and not in cold-recruited macrophages (*Figure 6—figure supplement 1A*). scRNA-seq distinguished one resident population of macrophages in control mice, and cold exposure led to the dramatic appearance of two new types of macrophages (*Figure 6B*). Cold exposure increased the number of MAC1 and MAC3 macrophage populations

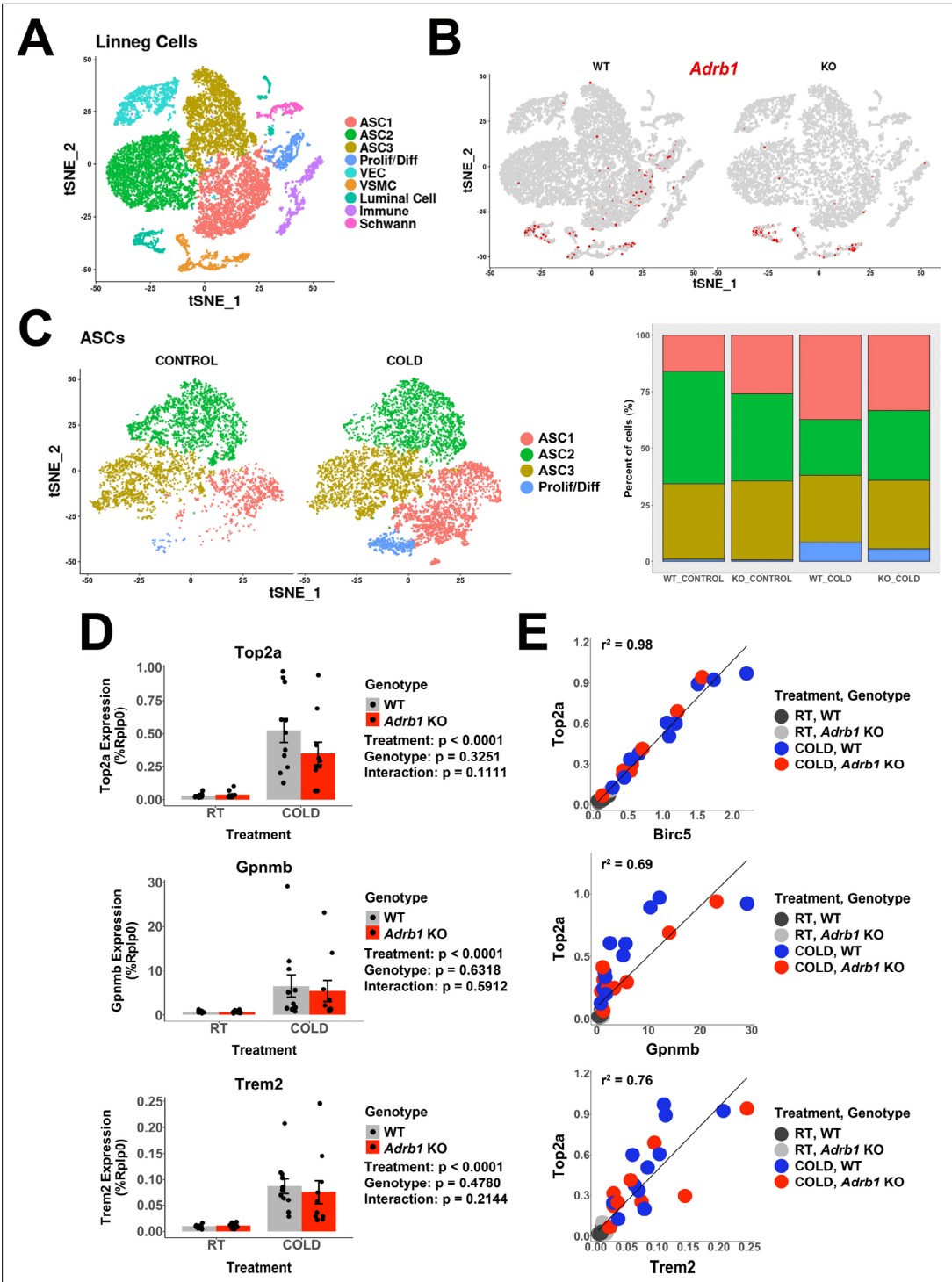

**Figure 5.** *Adrb1* is dispensable for cold-induced brown adipocyte neogeneisis. Related to *Table 1A*. (**A**) t-SNE plot of 18,332 Lin- cells from iBAT of WT (*Adrb1^fl/fl*) or *Adrb1* KO mice, either housed at room temperature or exposed to cold for 4 days. Clustering identified 9 cell types. ASC, adipose tissue stromal cell; VEC, vascular endothelial cell; VSMC, vascular smooth muscle cells; Prolif/Diff, proliferating/differentiating cells. DEGs that define these clusters are in **Supplementary file 3**. (**B**) t-SNE plot from (**A**), split into cells from WT or *Adrb1* KO animals. Colored cells have *Adrb1* expression > 0. (**C**) ASCs and Prolif/Diff cells from (**A**) reclustered, and split into cells from control or cold-exposed libraries. The bar graph shows the proportion of each cell type in the in the individual libraries. DEGs that define these clusters are in **Supplementary file 4**. (**D**) Quantitative PCR analysis of proliferation and immune cell activation genes in iBAT of WT or *Adrb1* KO mice maintained at room temperature or exposed to cold for four days (n=10–11 per condition; mean ± SE). p-values are from two-way ANOVA analysis of log transformed data. (**C**) Correlation of specific genes in the quantitative PCR data with *Top2a* expression by individual animal. r² values are displayed on the plot. See also **Figure 5—source data 1**.

*Figure 5 continued on next page*

*Figure 5 continued*

The online version of this article includes the following source data and figure supplement(s) for figure 5:

**Source data 1.** Quantitative PCR analysis of proliferation and immune cell activation genes in iBAT of WT or *Adrb1* KO mice maintained at room temperature or exposed to cold for 4 days (n=10–11 animals per condition).

**Figure supplement 1.** Role of ADRB subtypes in cold-induced iBAT neogenesis.

**Figure supplement 1—source data 1.** Quantification of the number of *Adrb1* molecules in *Nnat*+ cells between WT and *Adrb1* KO samples.

and reduced the number of MAC2 cells (*Figure 6B*). The MAC2 cluster had a resident macrophage profile, and expressed genes including *Folr2*, *F13a1*, *Cbr2*, *Pf4*, *Gas6*, and *Mrc1*. Some of these genes (*Folr2*, *Cbr2*, *Mrc1*) are also associated with an M2 anti-inflammatory macrophage phenotype. The largest cold-induced macrophage cluster, MAC1, had an anti-inflammatory and lipid-handling phenotype. MAC1 cells express the dead cell receptor *Trem2*, which functions to promote efferocytosis (*Figure 6—figure supplement 1C*). These cells also express the anti-inflammatory gene *Gpnmb*, and *Spp1*, which is known in other adipose depots as chemotactic for ASCs (*Figure 6—figure supplement 1C*). Apart from these markers, the top DEGs for MAC1 include genes involved in ECM remodeling (*Ctsb*, *Ctsd*, *Ctsl*, *Mmp12*), and lipid handling (*Fabp5*, *Lpl*, *Plin2*). These macrophages have a similar expression profile to that recently described during WAT remodeling (*Burl et al., 2018*; *Jaitin et al., 2019*). The second cold-induced macrophage cluster, MAC3, had lower expression of these MAC1 markers and higher expression of interferon-activated genes (*Ifi204*, *Ifi207*, *Ifi209*), several c-type lectin domain family 4 members (*Clec4d*, *Clec4e*, *Clec4n*), and many other interferon-regulated genes (*Irf7*, *Slfn2*, *Ifit3*, *Rsad2*, *Isg15*).

## Recruitment of cold-induced immune cells predicts the magnitude of progenitor proliferation and differentiation across experimental conditions

In this study, we performed 10 independent scRNA-seq experiments of >100,000 stromal and immune cells in which the magnitude of neogenesis was manipulated by temperature and ß-adrenergic receptor knockout and activation. Using these data, we assessed whether the induction of ASC1 proliferation and differentiation was associated with the recruitment of specific immune cell populations. We found that ASC1 proliferation and differentiation were strongly predicted by the magnitude of cold-induced recruitment of cDC1 (DEND1; $r^2$=0.74), moDC (DEND2; $r^2$=0.91), and lipid-handling macrophages (MAC1, $r^2$=0.87) across these diverse experimental conditions (*Figure 6C*

**Table 1.** Analysis of β-adrenergic receptor manipulation by scRNA-seq.

(A) Cell counts and calculated percentages for proliferating/differentiating cells (Prolif/Diff) and ASCs in the *Adrb1* KO single-cell libraries. p-values were calculated by chi-squared analysis between CONTROL and COLD libraries for the two genotypes. ***p<1E-5. (B) Cell counts and calculated percentages for Prolif/Diff cells and ASCs in the CONTROL and CL-treated single-cell libraries. p-values were calculated by chi-squared analysis between CONTROL and CL libraries. ***p<1E-5.

| A | Library | Number of Prolif/Diff Cells | Number of ASCs | Percent of Prolif/Diff Cells out of Total ASCs (%) |
|---|---|---|---|---|
| | CONTROL, WT | 31 | 3,695 | 0.84 |
| | COLD, WT | 437 | 5,118 | 8.54*** |
| | CONTROL, *Adrb1* KO | 9 | 1,543 | 0.58 |
| | COLD, *Adrb1* KO | 171 | 3,063 | 5.58*** |

| B | Library | Number of Prolif/Diff Cells | Number of ASCs | Percent of Prolif/Diff Cells out of Total ASCs (%) |
|---|---|---|---|---|
| | CONTROL | 38 | 7,234 | 0.53 |
| | CL | 1,526 | 11,275 | 13.5*** |

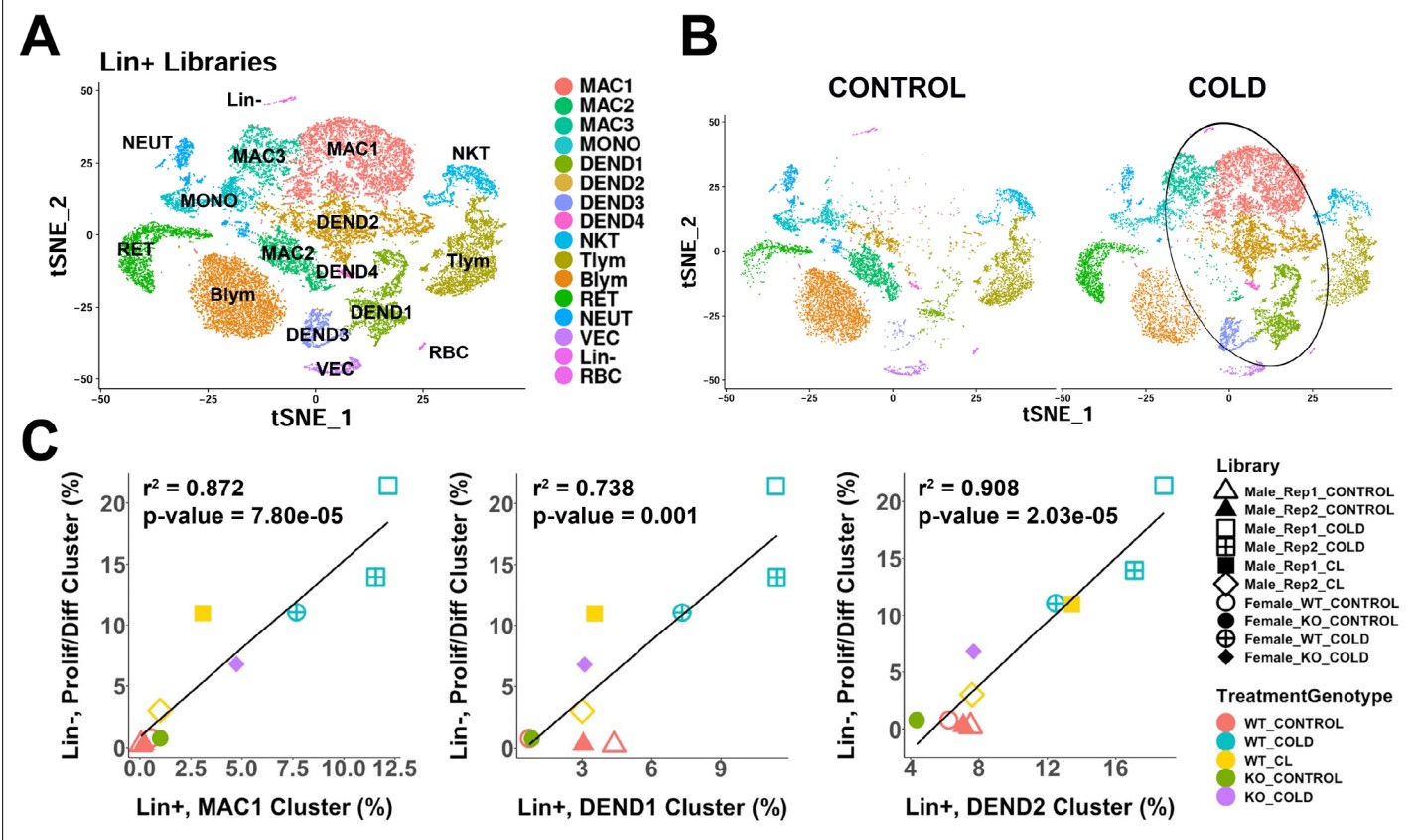

**Figure 6.** scRNA-seq analysis of Lin+ immune cells from iBAT of control or cold-exposed mice. (**A**) t-SNE plot of 25,344 lineage marker positive (Lin+) cells from iBAT of control mice and mice exposed to cold for four days. Clustering applied to the t-SNE plot identified 16 clusters, highlighted in different colors. MAC, macrophage; MONO, monocyte; DEND, dendritic cell; NKT, natural killer T-cell; Tlym, T lymphocyte; Blym, B lymphocyte; RET, reticulocyte; NEUT, neutrophil; VEC, vascular endothelial cell; Lin-, lineage-negative cells; RBC, red blood cell. Cell types were determined by DEGs within each cluster. DEGs that define these clusters are in ***Supplementary file 7***. (**B**) t-SNE plot from (**A**) split into the cells from control (CONTROL) and cold-exposed (COLD) mice. Circle highlights cold-induced cell clusters. (**C**) Lin+ or Lin- libraries from the 10 single-cell experiments presented in this paper were visualized and clustered in one t-SNE plot and the proportion of each immune cell cluster in the Lin+ libraries were correlated with the proportion of the proliferating/differentiating cluster in the Lin- libraries. Point shapes correspond to individual libraries. Point color corresponds to the combination of treatment (CONTROL, COLD, CL) and the genotype (WT, KO). r² values and p-values are displayed on the plot. p-values were calculated using the Pearson's product-moment correlation.

The online version of this article includes the following figure supplement(s) for figure 6:

**Figure supplement 1.** Analysis of gene expression markers from single-cell analysis of iBAT dendritic cells and lipid-handling macrophages.

and *Figure 6—figure supplement 1D*). Importantly, this highly significant relationship held even when basal values were excluded from the analysis.

## Macrophages and dendritic cells interact with proliferating ASCs in known sites of neogenesis

The data above suggest a scenario in which metabolic stress in brown adipocytes leads to immune cell recruitment, niche formation, and brown adipocyte neogenesis, similar to that observed for brown adipocyte neogenesis induced by ADRB3 activation in WAT (*Lee and Granneman, 2012*; *Lee et al., 2016*; *Lee et al., 2012*). We previously reported that cold-induced neogenesis in mice is concentrated near tissue borders, although with intense drug induction, neogenesis can expand into the parenchyma (*Lee et al., 2015*). Furthermore, analysis of adipogenesis induced by ADRB3 activation in white adipose tissue demonstrated a close interaction of immune cells (mainly macrophages) and progenitor cells at sites of dead adipocyte removal. Therefore, we next examined the distribution of immune cells relative to tissue architecture and proximity to activated progenitors, utilizing informative cell markers extracted from scRNA-seq analysis.

MAC2 are the predominant macrophage subtype of control (RT) mice and their levels sharply declined during cold exposure (*Figure 6B*). MAC2 express F4/80 (*Adgre1*) and MHCII (*H2-Ab1*), have a compact morphology, and are evenly distributed throughout the parenchyma (*Figure 7A*). As expected, few stromal or immune cells of control mice were positive for the proliferation marker MKI67 (*Figure 7A*). Histological analysis of iBAT at the peak for cold-induced proliferation demonstrated the dramatic appearance of GPNMB+ MAC1 cells and dendritic cells labeled with MHCII antibodies (noting that MHCII+ MAC2 are nearly absent in iBAT of cold-exposed mice; *Figure 7A*). Interestingly, nearly all MAC1 and dendritic cells were present in clusters in two distinct locations: within 50 micrometers of the tissue border or near acellular vacancies in the parenchyma. As expected, low resolution imaging identified numerous MKI67+ cells in sections of cold-exposed iBAT, indicative on ongoing proliferation (*Figure 7A*).

High-resolution confocal imaging of the tissue border region demonstrated that nearly all actively proliferating ASC1 (MKI67+ PDGFRA+) cells were in close proximity to F4/80+ immune cells (*Figure 7B* and *Appendix 1—Video 1*). According to our single-cell data, F4/80 (*Adgre1*) is expressed in macrophage clusters and DEND2 (moDCs). ASC1 extend cellular processes that have the potential to probe the tissue microenvironment (*Lee et al., 2015*). 3-D confocal microscopy of cold-exposed iBAT indicates that ASC1 cellular processes extend toward and contact F4/80+ cells (*Figure 7B* and *Appendix 1—Video 1*).

We also observed GPNMB+ MAC1 and MHCII+ dendritic cells surrounding vacancies lacking cellular autofluorescence and PLIN1, a lipid droplet marker of live adipocytes (*Figure 7C* and *Appendix 1—Video 2*). Such structures are the hallmark of adipocytes undergoing death and replacement (*Lee et al., 2013a*; *Murano et al., 2013*). Indeed, all PLIN1 negative vacancies had a uniform diameter of approximately 20 µm which closely matches the size of adjacent BAs, and many retained remnants of intracellular lipid droplets in cryosections (*Figure 7C*, *Appendix 1—Video 2*, and *Figure 7—figure supplement 1*). Thus, the vacancies appear to be sites of efferocytosis by GPNMB+ MAC1 and MHCII+ dendritic cells. Importantly, these sites contained actively proliferating (MKI67+) ASC1 (PDGFRA+) near immune cells (F4/80+) (*Figure 7D*).

## Adipogenic niches are dynamic

We estimate that at the peak of neogenesis there are hundreds of adipogenic niches in various stages of initiation, proliferation and differentiation. We hypothesized that smFISH analysis of state-dependent markers (e.g. *Top2a* and *Nnat*) might provide clues regarding the cellular basis of adipogenic niche formation and resolution. For this purpose, we used smFISH to systematically assess the proximity of immune cells (*H2-Ab1*) with proliferating (*Dcn+ Top2a+*) and differentiating progenitors (*Nnat+*), and considered neighbors within 20 µm (the thickness of cryosections) of the reference cell type. Analysis by machine learning classification and automated measurement yielded similar results. As expected from immunofluorescence analysis (*Figure 8A–B*), proliferating progenitors and differentiating adipocytes were nearest neighbors (NN) of themselves, reflecting clustering within adipogenic niches. Furthermore, proliferating progenitors were closely associated with *H2-Ab1+* dendritic cells (*Figure 8A*) and this close association declined by four- to five-fold as progenitors differentiated into BAs (*Figure 8B*). These data indicate that progenitor proliferation occurs in close proximity to recruited immune cells, which depart as progenitors differentiate into BAs.

smFISH provides a tissue 'snapshot' suggesting dynamic trafficking and interactions among progenitors and phagocytes in tissue renewal and expansion. To further establish this concept, we mapped the relative positions of actively proliferating ASC1 and newly-differentiated BAs relative to dendritic cells in the tissue microenvironment. For this purpose, PDGFRA+ progenitors were genetically labeled using Pdgfra-CreER$^{T2}$ x LSL-tdTomato mice prior to cold exposure (*Figure 8—figure supplement 1A*), then we flash-tagged proliferating cells with 5-ethynyl-2'-deoxyuridine (EdU) and monitored their positions relative to immune cells immediately after tagging (on the third day), or 2 days later when most cells had differentiated into BAs. This analysis indicated that MHCII+ DC were three times more likely to be with 10 micrometers of a dividing preadipocyte versus a divided/differentiated brown adipocyte (27.3% vs 9.35%, p=0.00013, n=267 cells, 2–3 mice) (*Figure 8—figure supplement 1A*). We confirmed spatial relationships among actively dividing and newly differentiated progenitors by immunofluorescence (*Figure 8—figure supplement 1B*). In this experiment, we also located a limited number of progenitors that divided on day 3, yet failed to differentiate by day 5, and

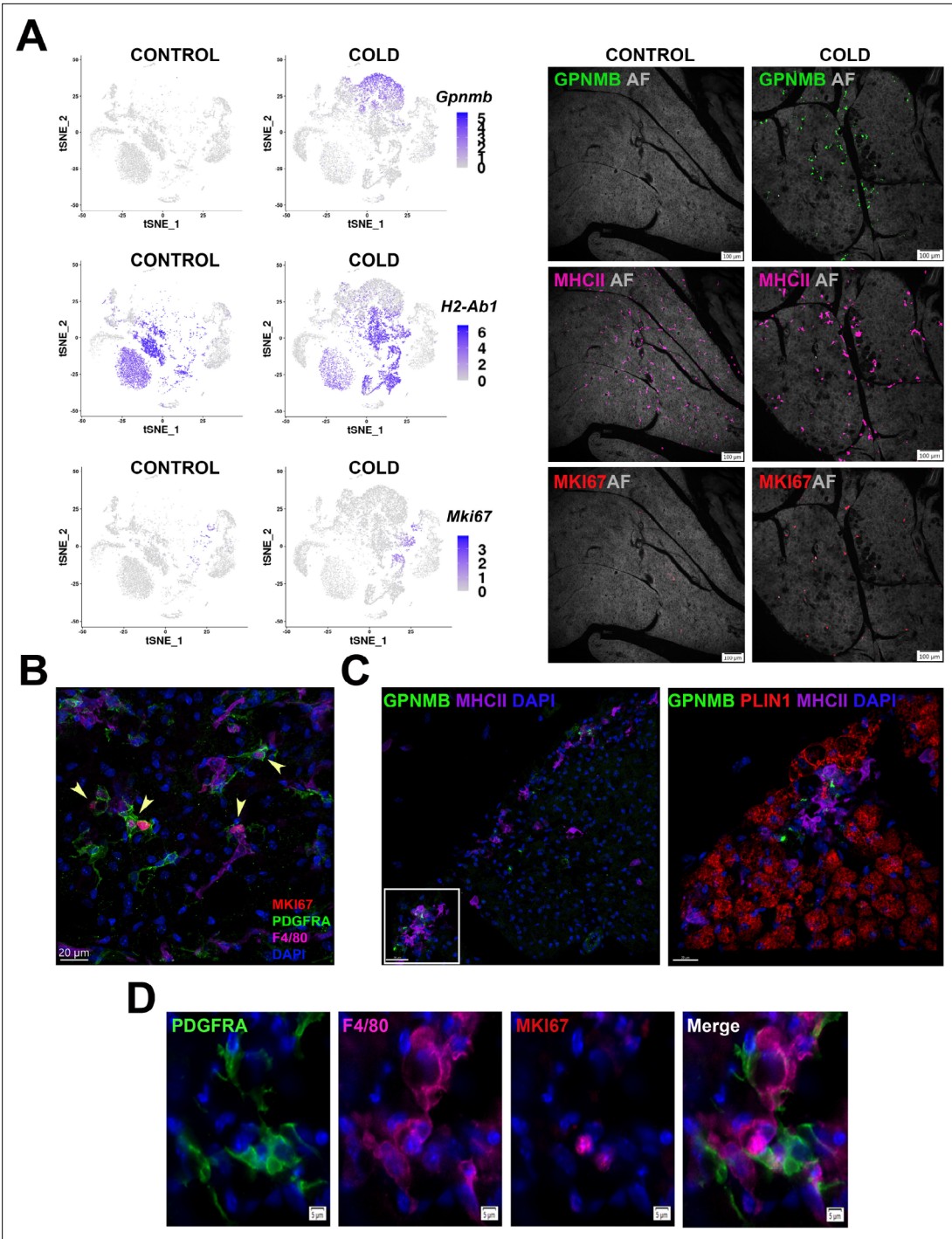

**Figure 7.** Cold-induced iBAT neogenesis involves close interaction between immune cells and proliferating ASCs. (**A**) Low-magnification images of iBAT fixed-frozen sections from control and cold-exposed mice stained for GPNMB (green), MHCII (pink), and MKI67 (red), along with their corresponding log2 gene expression in Lin+ scRNA-seq data. Lin+ t-SNE plots were split between cells from control and cold-exposed mice. Tissue images for antibody staining show the same tissue area for each treatment. Background for each image is autofluorescence (AF) in grey scale. Scale bar, 100 µm. (**B**) iBAT fixed-frozen sections from cold-exposed mice stained with antibodies for MKI67 (red), PDGFRA (green), and F4/80 (pink). Nuclei were counterstained with DAPI. Arrows indicate PDGFRA+ MKI67+ cells. Scale bar, 20 µm. 3D rendering presented in *Appendix 1—Video 1*. (**C**) iBAT fixed-frozen sections from cold-exposed mice. Left image displays antibody staining for GPNMB (green) and MHCII (purple). Right image is a magnified view of the boxed region from left, displaying staining for GPNMB (green), PLIN1 (red), and MHCII (purple). Nuclei were counterstained with DAPI. Scale bars, 30 µm and 20 µm. 3D rendering presented in *Appendix 1—Video 2*. (**D**) iBAT fixed-frozen sections from cold-exposed mice stained for PDGFRA (green), F4/80 (pink), and MKI67 (red). Nuclei were counterstained with DAPI. Scale bar, 5 µm.

*Figure 7 continued on next page*

*Figure 7 continued*

The online version of this article includes the following figure supplement(s) for figure 7:

**Figure supplement 1.** Macrophages surround cellular vacancies containing lipid remnants.

found that nearly all of these cells maintained close contact with MHCII+ cells in the tissue microenvironment (*Figure 8—figure supplement 1B*). Taken together, these observations indicate that ASC1 and recruited immune cells comprise a cellular niche for progenitor proliferation and that immune cells depart the niche as progenitors differentiate into BAs.

## Investigating potential cell-cell communication in the adipogenic niche

The close proximity of the cold-activated ASC1 and immune cell subtypes suggests cell-to-cell communication orchestrates the neogenic process. To gain insights into potential ligand-receptor interactions between immune and progenitor cells, we interrogated our scRNA-seq data for ligand-receptor pairs using published databases and publicly available computational programs (*Browaeys et al., 2020*; *Efremova et al., 2020*; *Skelly et al., 2018*). All pairs identified by these methods were manually examined for ligand-receptor specificity in our dataset. We focused on ligand-receptor pairs between immune cells (MAC1 and DEND1-4) and cold-responsive ASC1. This analysis identified several ligand-receptors pairs that could contribute to adipocyte progenitor recruitment and

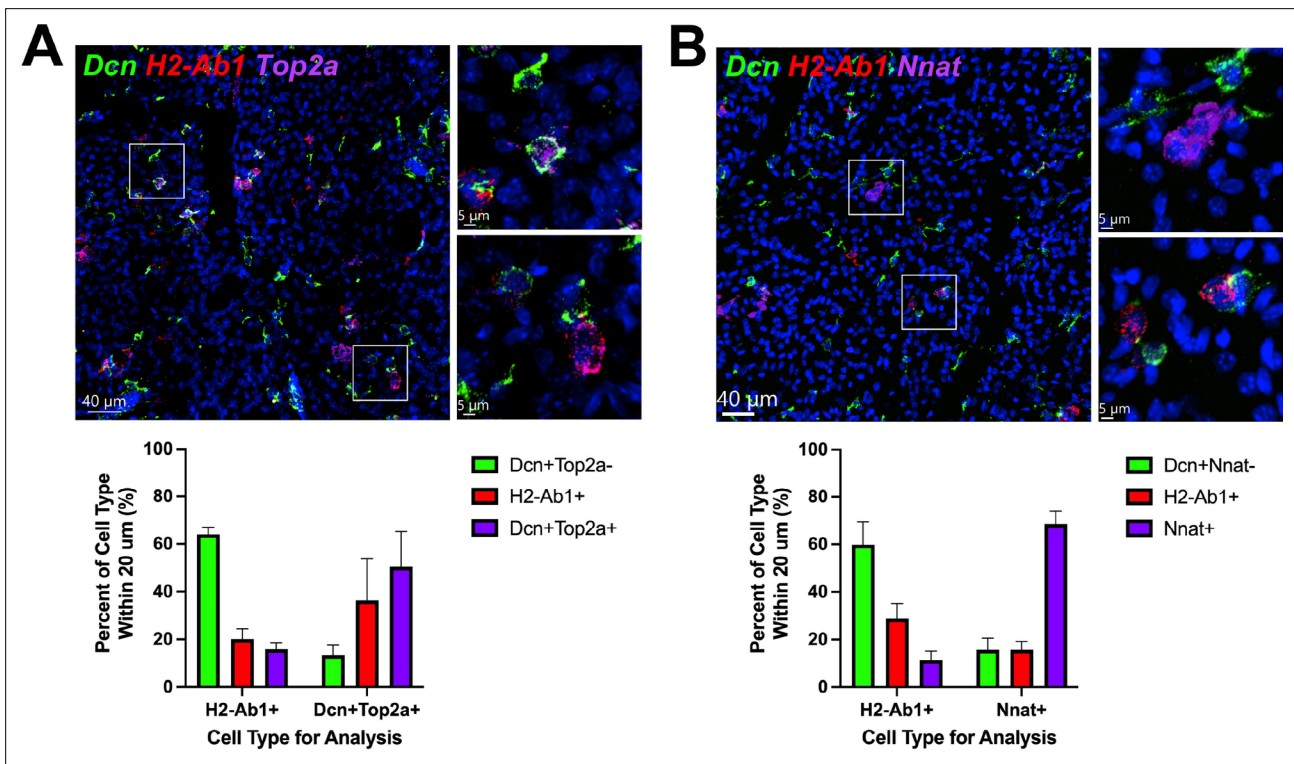

**Figure 8.** ASCs and immune cells comprise a dynamic cellular niche. (A, top) Representative images of fixed frozen mouse cold-exposed iBAT. Tissue was stained with smFISH probes *Dcn* (green), *H2-Ab1* (red), and *Top2a* (pink). Nuclei were counterstained with DAPI. Images on the right are a magnified view of the boxed regions on the left. Scale bars, 40 µm and 5 µm. (A, bottom) Quantification of cell types within 20 µm around either an *H2-Ab1*+ cell or a *Dcn+ Top2a*+ cell (n=3 animals; >100 cells/mouse; mean ± SD). (**B**) Representative images of fixed frozen mouse cold-exposed iBAT. Tissue was stained with smFISH probes *Dcn* (green), *H2-Ab1* (red), and *Nnat* (pink). Nuclei were counterstained with DAPI. Images on the right are a magnified view of the boxed regions on the left. Scale bars, 40 µm and 5 µm. (B, bottom) Quantification of cell types within 20 µm around either an *H2-Ab1*+ cell or a *Nnat*+ cell (n=3 mice; >100 cells/mouse; mean ± SD). See also *Figure 8—source data 1*, *Figure 8—source data 2*.

The online version of this article includes the following source data and figure supplement(s) for figure 8:

**Source data 1.** Quantification of the nearest neighbor (20 µM) to either a *H2-Ab1*+, *Dcn*+, or *Dcn+ Top2a*+ cell in the iBAT of cold-exposed mice.

**Source data 2.** Quantification of the nearest neighbor (20 µM) to either a *H2-Ab1*+, *Dcn*+, or *Nnat*+ cell in the iBAT of cold-exposed mice.

**Figure supplement 1.** Flash-labeling of proliferating cells and proximity of MHCII+ dendritic cells.

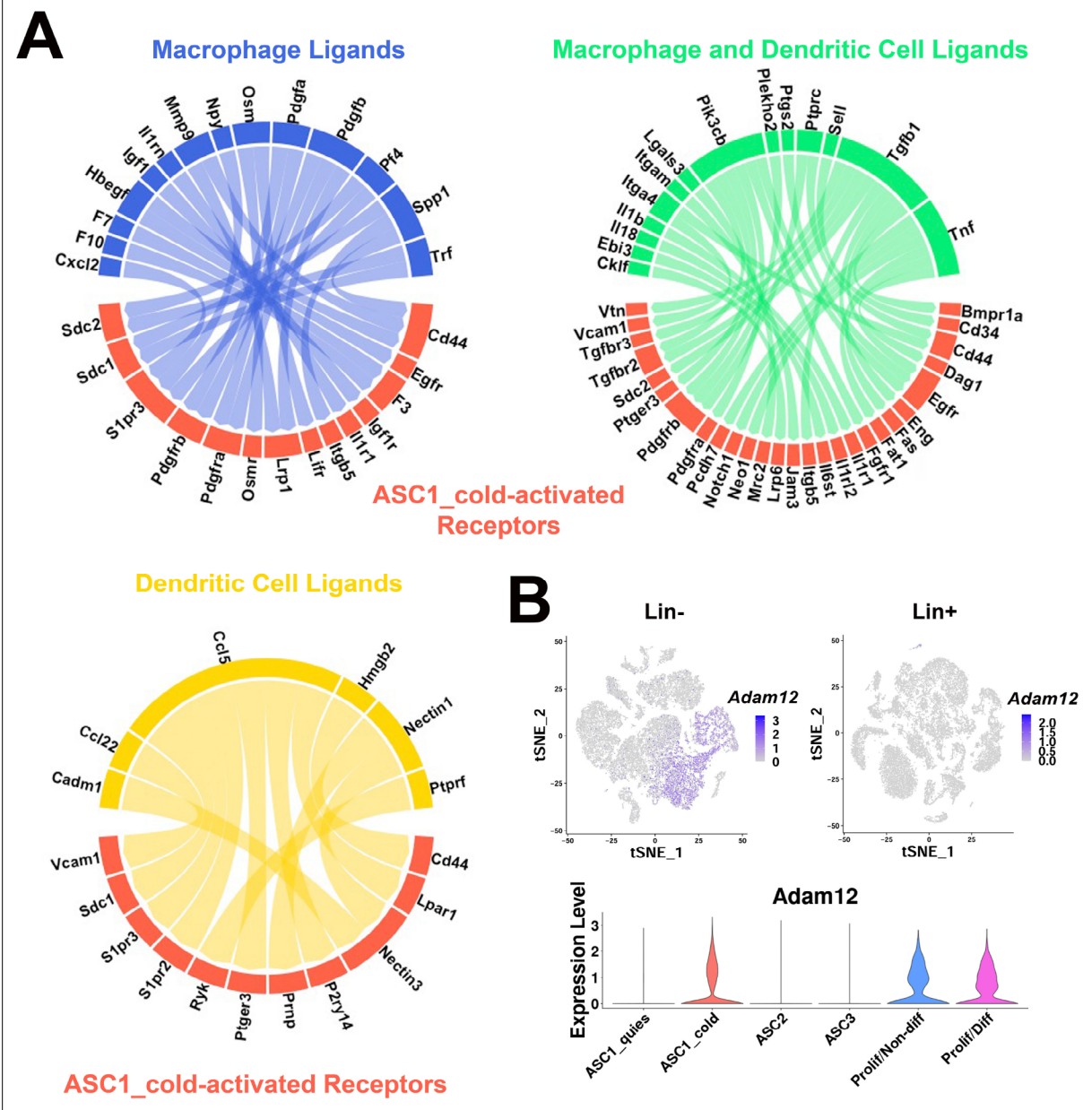

**Figure 9.** scRNA-seq data reveal potential ligand-receptor pairs for ASC-immune cell crosstalk. (**A**) Circos plots of ligand-receptor pairs, where cold-activated ASC1 cells expressed the receptor and dendritic cells and/or macrophages expressed the ligand. Ligand-receptor pairs were identified from the computational programs CellPhoneDB and NicheNet, and additional ligand-receptor databases. Phenotype specificity was confirmed using expression patterns in t-SNE plots. (**B**) t-SNE plots displaying the log2 expression pattern levels for *Adam12* corresponding to Lin- data (*Figure 2A*), Lin+ data (*Figure 6A*), and ASC re-clustered data (*Figure 2C*), respectively.

differentiation (*Figure 9A*). Osteopontin (*Spp1*), a DEG in the cold-induced MAC1 cluster (*Figure 6—figure supplement 1C*), is known to be chemotactic for ASCs in other adipose depots (*Lee et al., 2016*; *Lee et al., 2012*). Cd44 molecule (*Cd44*) and Syndecan 1 (*Sdc1*) were identified as receptors for *Spp1* in our queried databases (*Figure 9A*). *Cd44* has already been identified as a marker of activated adipocyte progenitors in other depots (*Lee et al., 2016*; *Lee et al., 2013a*; *Lee et al., 2014*). *Sdc1* has been categorized as a regulator of adipogenesis (*Gougoula et al., 2019*; *Yu et al., 2020*; *Zaragosi et al., 2015*), and its expression was highly specific to the cold-activated ASC1 cells.

Cold-responsive immune cells and ASC1 also selectively upregulated genes involved in ECM remodeling, including several metalloproteinases that are known to activate latent growth factors in the microenvironment. In this regard, cold exposure greatly upregulated ASC1 expression of

the disintegrin and metalloproteinase ADAM metallopeptidase domain 12 (*Adam12*; *Figure 9B*). ADAM12 is multifunctional sheddase that could promote ASC1 proliferation and differentiation by multiple mechanisms, including local release of active HB-EGF and augmentation of IGF1 signaling via inactivation of IGFBP3 (*Taylor et al., 2014*; *Loechel et al., 2000*). Indeed, ADAM12 promotes adipogenesis in vitro, and ectopic overexpression of ADAM12 in muscle is sufficient to induce fat pad formation in vivo (*Coles et al., 2018*; *Kawaguchi et al., 2002*).

## Discussion

Neogenesis is an important adaptive mechanism for expanding the thermogenic capacity of BAT during sustained cold exposure (*Bukowiecki et al., 1982*; *Bukowiecki et al., 1986*; *Foster and Frydman, 1978*; *Nedergaard, 1982*; *Nedergaard et al., 2019*), yet the cellular mechanisms involved are poorly understood. It is well established that cold-induced neogenesis is mediated by sympathetic nerve activity, can be mimicked by systemic infusions of NE (*Géloën et al., 1992*; *Lee et al., 2015*), and global knockout of ADRB1 prevents induction of neogenesis by NE infusion in vivo (*Lee et al., 2015*), supporting the hypothesis that activation of progenitor ADRB1 mediates cold-induced BA neogenesis. However, this simple scenario does not readily account for the observations that neogenesis (1) commences after 2–3 days of cold exposure (*Bukowiecki et al., 1986*; *Hunt and Hunt, 1967*; *Lee et al., 2015*), well after cold activation of classic β-adrenergic/PKA signaling, (2) occurs in specific locations in iBAT (*Lee et al., 2015*), and (3) correlates with recruitment of myeloid cells to the tissue.

To address these phenomena, we performed scRNA-seq and fate mapping in situ to identify the source of cold-induced BAs among the ASC subtypes present in the tissue. We determined ASC *Adrb1* is not necessary, whereas activation of adipocyte ADRB3 is sufficient to induce BA neogenesis in vivo. We confirmed that iBAT neogenesis begins only after 2–3 days of cold exposure, and further determined that the onset and magnitude of neogenesis in individual mice was tightly correlated with specific immune cell recruitment, but not the acute or chronic induction of genes known to be regulated by the β-adrenergic/PKA signaling pathway. Together, these results indicate that cold-induced neogenesis is largely an indirect response to chronic activation of BAs that ultimately serves to increase thermogenic capacity to match thermogenic demand. From a physiological perspective, this scenario implies the existence of mechanisms that report metabolic stress, which is detected and transduced into targeted adaptive hyperplasia.

### BAT cell type heterogeneity

Our scRNA-seq data highlights the diversity of *Pdgfra+* stromal cells in BAT and indicate that *Pdgfra* expression most clearly defines ASCs, whereas *Cd34*, *Ly6a*, and *Pdgfrb* are present in additional stromal vascular subtypes. We identified three populations of ASCs (ASC1, ASC2, ASC3). Importantly, the ASC subtypes can be distinguished by differential expression of genes that are involved in ECM production and signaling by members of the TGF-β superfamily. Histological evidence from other adipose depots determined a spatial distinction among the main ASC subtypes (*Rondini and Granneman, 2020*; *Merrick et al., 2019*). ASC2 cells (PDGFRA+ DPP4+ Pi16+) were restricted mainly to the fascia layer surrounding tissue, while ASC1/ASC3 cells (PDGFRA+ DPP4- Pi16-) were found throughout the parenchyma body and in areas surrounding vessels. We observed a similar spatial distinction using smFISH, with ASC3 cells surrounding vessels, ASC2 cells present in the fascia that encases the tissue lobes, and ASC1 cells residing in the tissue parenchyma. These distinctions, however, were not absolute, as we observed cells expressing *Pi16* and *Gdf10* on some large vessels, and interstitial ASC1 cells closely associated with capillaries. Together, these data indicate that the ASC subtypes serve distinct functions in the tissue microenvironment. We note in this regard that both ASC1 and ASC2 can be differentiated into adipocytes in vitro by strong chemical inducers (unpublished); thus, in vitro differentiation potential should not be used to infer in vivo contributions to new fat cell formation.

Eight of our single-cell libraries were prepared from iBAT cells from female mice, while 12 were prepared from iBAT cells from male mice. This gave us the opportunity to compare and query whether there are sex-specific iBAT stromal cell subtypes. Interestingly, we found a small cell cluster (1.4–6.4% of cells per library) in the female Lin- libraries (*Adrb1* KO and WT) that we labeled Luminal Cells. This Luminal Cell cluster was distinct from other Lin- cell types, was reproduced in every female Lin- library,

and the top DEGs that defined this cluster were not found in the male libraries. DEGs for this cluster include kertain genes (*Krt18, Krt19, Krt8, Krt7*), WAP four-disulfide core domain genes (*Wfdc18, Wfdc2*), and epithelial cell adhesion molecule (*Epcam*). This profile is very similar to a mouse mammary luminal cell phenotype, as profiled in the *Tabula Muris* atlas (*Schaum et al., 2018*). Although the function of this cell phenotype in BAT is unclear, we note that this cluster selectively expresses the progesterone receptor (*Pgr*), and that the effect of female hormones could be driving this cell phenotype (*Kaikaew et al., 2021*).

## Origin of cold-induced BA cells

The identity of the direct precursors of new brown adipocytes remains a highly debated topic (*Lee et al., 2015*; *Lee et al., 2012*; *Shamsi et al., 2021*). Early fate mapping by Bukoweicki et al. strongly indicated that brown adipocytes arise from interstitial cells that are transiently recruited in the first few days of cold stress (*Bukowiecki et al., 1982*; *Bukowiecki et al., 1986*; *Géloën et al., 1988*). Subsequently, genetic tracing and fate mapping studies established that brown adipocytes arise from an interstitial stromal cell population that express PDGFRA in brown adipose tissue (*Cattaneo et al., 2020*; *Lee et al., 2015*; *Sun et al., 2020*). In further support of this conclusion, we repeated the lineage tracing experiments described in *Lee et al., 2015* using an independent Pdgfra-CreER$^{T2}$ x LSL-tdTomato mouse model and found that 100% (258/258, n=5 mice) of new brown adipocytes were derived from the proliferation of cells with a recent history of *Pdgfra* expression.

Although *Myf5* expression has been linked to development of the BA lineage (*Lee et al., 2015*; *Seale et al., 2008*), it was not present/expressed in any adult stromal cell subtype. Rather, our scRNA-seq data clearly demonstrate that *Pdgfra+* ASC1 cells are the immediate progenitors of cold-induced BAs in adult iBAT. Thus, proliferating/differentiating cells expressed ASC1 markers, but not markers of ASC2 or ASC3. In addition, smFISH analysis demonstrated that a high proportion of actively differentiating cells that express *Nnat* co-express the ASC1 marker *Bmper*. Some proliferating ASC1 cells proceeded along an adipogenic trajectory, upregulating markers such as *Car3*, *Adipoq*, and *Plin1*, while some proliferating cells did not. These non-differentiating cells could function to maintain populations of ASC1 cells in iBAT. What determines these different cells fates is not known, but our data suggests that microenvironmental factors, such as proximity to recruited immune cells, could play a role.

While this work was in progress, Shamsi, et al. published scRNA-seq data of iBAT from cold-exposed mice suggesting that *Trpv1+* vascular smooth muscle cells (VSMCs) can give rise to BAs (*Shamsi et al., 2021*). Although *Trpv1*-expressing VSMCs are present in our dataset, these cells were not linked to an adipogenic trajectory. In contrast, we only identified adipogenic differentiation from *Pdgfra+* ASCs by scRNA-seq and genetic tracing. We believe that key differences in experimental design might have contributed to the differences in results and interpretation. First, our dataset profiled mice at the peak in progenitor proliferation (4 days of cold), while the Shamsi, et al. dataset profiled mice after two and seven days of cold exposure in which little neogenesis is observed (*Lee et al., 2015*; *Shamsi et al., 2021*). Notably, no proliferating cells are present in the *Shamsi et al., 2021* scRNA-seq data. In contrast, we observe robust (~8–21%) proliferation and differentiation of ASC1 in multiple scRNA-seq experiments, and have confirmed this adipogenic trajectory in situ by inducible genetic lineage tracing, chemical fate mapping, and smFISH. Second, the datasets contained drastically different proportions of cell types, despite coming from the same tissue. Adipocytes and ASCs make up only an estimated ~15% of the SVF cells in the Shamsi, et al. data (est.~3,500 cells), compared to ~68.5% of Lin- cells in the present data (19,659 cells). This difference likely involves cell isolation methodology, such as digestion reagents and centrifugation speed (unpublished observations).

In view of potential differences in the efficiency and recovery of isolated stromal cells, it was important to establish the cold-induced adipogenic trajectory in situ using highly sensitive smFISH. High resolution confocal microscopy of informative mRNA markers established the presence of cold-induced adipogenic niches containing a range of adipogenic states of ASC1 from quiescence to active proliferation, and from early to late differentiation. Spatial mapping of this trajectory indicates that adipogenic niches involve close (micron scale) cellular communication in tissue volumes less than 0.0001 cubic millimeter, consistent with the estimated distance of cytokine signaling in situ (*Oyler-Yaniv et al., 2017*).

## Cold-induced neogenesis occurs in distinct locations and involves cell turnover and immune cell recruitment

Previous FACS analysis of iBAT neogenesis identified a population of F4/80+ proliferating cells that are recruited after cold stress (*Lee et al., 2015*). Current scRNA-seq data provide new insights into the composition and location of cold-induced immune cell recruitment. Specifically, cold exposure led to the reorganization of immune cell composition of iBAT that included reduction of resident MAC2 cells and striking recruitment of phagocytic macrophage and dendritic cell subtypes.

Macrophage involvement in adipogenesis has been well documented in white adipose depots (*Lee et al., 2016*; *Lee et al., 2013a*), but remains poorly described in brown adipose tissue. While both macrophage clusters MAC1 and MAC3 were profiled specifically in libraries from cold-exposed mice (*Figure 5B*), the MAC1 transcriptional profile suggests these cells are involved in the adipogenic process. Top DEGs for MAC1 included genes involved in extracellular matrix remodeling (*Ctsd*, *Ctsb*, *Ctss*), lipid handling (*Fabp5*, *Plin2*, *Lpl*), and chemoattractant activity (*Spp1*). MAC1 cells also expressed high levels of *Gpnmb* and *Trem2* (*Figure 6—figure supplement 1C*). A similar macrophage gene expression profile was observed in single-cell data from mouse white adipose tissue, and, under certain conditions, corresponds to macrophages involved in adipogenesis (*Burl et al., 2018*; *Jaitin et al., 2019*). TREM2+ macrophages have been well reported in the liver, where they are protective against non-alcoholic fatty liver disease (*Dou et al., 2016*; *Hou et al., 2021*; *Jaitin et al., 2019*; *Xiong et al., 2019*). TREM2 itself has been documented in the brain primarily as a dead cell receptor involved in efferocytosis (*Filipello et al., 2018*; *Hu et al., 2021*; *Kleinberger et al., 2014*; *Krasemann et al., 2017*; *Nugent et al., 2020*). In gonadal WAT, dying adipocytes recruit macrophages which in turn recruit ASCs for proliferation and differentiation (*Lee et al., 2016*; *Lee et al., 2013a*). The presence of *Trem2*+ macrophages in our iBAT dataset, along with histology, indicate that cold exposure induces localized cellular turnover that involves immune cell recruitment.

Surprisingly, cold exposure also recruited several dendritic cell subtypes, including conventional and monocyte-derived subtypes, and lesser numbers of mature/migratory and plasmacytoid dendritic cells. Moreover, dendritic cells were the primary proliferating immune cell type in the lineage positive cell fraction. Dendritic cells are typically considered antigen-presenting cells that modulate adaptive and innate immune responses. However, roles of dendritic cells in non-immune signaling functions have been recognized, such as tissue homeostasis and wound healing (*Anzai et al., 2012*). These dendritic cells likely function in a more homeostatic role, as no overt lymphocyte activation was observed. Because monocyte-derived dendritic cells express *Adgre1* (encoding F4/80), we conclude that the proliferating F4/80+ cells we previously identified by FACS were primarily monocyte-derived dendritic cells.

Proliferation and differentiation were highly correlated with the recruitment of monocyte-derived dendritic cells and lipid-handling macrophages, so it was critical to locate these cellular phenotypes in the BAT microenvironment. Histological analysis demonstrated that cold exposure recruited immune cells to specific tissue subdomains that have been previously associated with neogenesis in iBAT (*Lee et al., 2015*) and WAT (*Lee et al., 2013a*; *Lee et al., 2012*). These included a region near the border of iBAT lobes and along the interface between iBAT and suprascapular WAT (*Lee et al., 2015*). The mechanisms for recruitment to this tissue subdomain are not known, but we note that ASC2, which comprises the fascia, is the dominant ASC subtype in this region. ASC2 exhibit an immunosurveillance phenotype and cold exposure upregulated expression of genes involved in response to stress and immune defense (GO:0006950 and GO:0006955, FDR *P*<10E-10) in these cells. In view of the fact that parenchymal BAs are physically interconnected by gap junctions (*Burke et al., 2014*; *Revel et al., 1971*; *Zhu et al., 2016*), tissue neogenesis might proceed more effectively near tissue borders. Thus, we suggest that ASC2 are well positioned to sense and communicate metabolic stress, and position nascent niches for tissue expansion.

Histological analysis also indicated that chronic cold stress induces parenchymal cell turnover, involving removal of defective BAs and their precise replacement by ASC1 progenitors. We found that cold stress led to the appearance of cellular vacancies within the parenchyma that were the size of BAs and often contained the coalesced remnants of lipid droplets that lacked PLIN1, a marker of live adipocytes. These structures were surrounded by GPNMB+ and MHCII+ macrophage and dendritic cells indicative of ongoing efferocytosis, and proliferating ASC1 cells involved in cell replacement. While mechanistic details are yet to be unraveled, our data suggest that lipid-handling MAC1 cells are

recruited to distressed BAs. MAC1 express high levels of active mitogens, like *Igf1* and *Pdgfa*, which have cognate receptors on ASC1 cells. In addition, MAC1 cells express high levels of matrix remodeling proteases that are known to release latent growth factors, like HB-EGF, and modulate the local action of growth factor signaling by inactivating inhibitory growth factor binding proteins (*Loechel et al., 2000*; *Nakamura et al., 2020*).

Dendritic cells occupied the same niche with macrophages and proliferating PDGFRA+ cells. Whether dendritic cells are recruited in tandem by signals from distressed adipocytes or are recruited to the adipogenic niche by signals from recruited macrophages is presently unclear. Our data suggests that MHCII+ dendritic cells are more heavily recruited to the iBAT periphery, whereas GPNMB+ MAC1

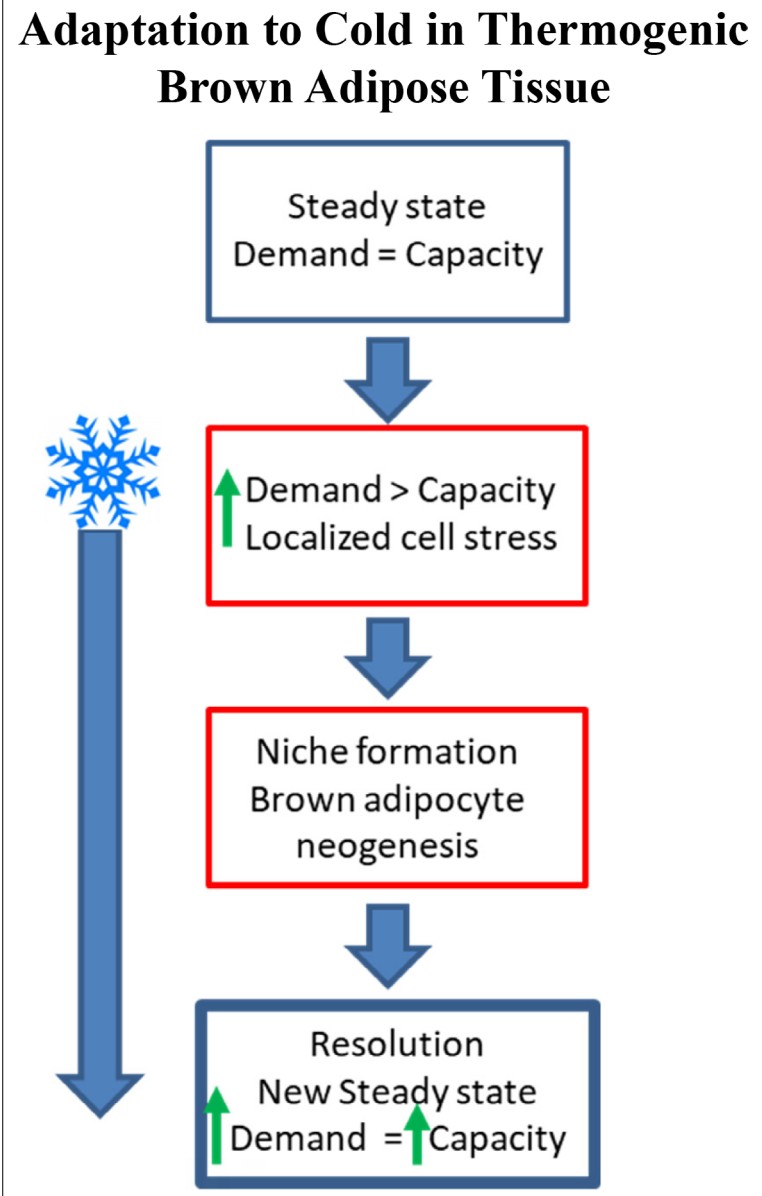

**Figure 10.** Schematic demonstrating adaptation to cold-induced stress in brown adipose tissue (non-shivering thermogenesis). Under steady state conditions, the capacity for thermogenesis is in equilibrium with the demand for thermoregulation. As the demand for heat production increases (i.e. cold exposure) BAT can increase thermogenic capacity (upregulate UCP1, mitochondrial content); however, continued cold exposure exceeds the capacity of existing adipocytes (non-shivering thermogenesis) and muscle (shivering thermogenesis) to regulate body temperature, resulting in brown adipocyte niche formation and neogenesis until a new steady state level is reached.

are associated with efferocytosis in the iBAT parenchyma. It thus seems likely that the exact composition of adipogenic niches varies among microdomains. Fate mapping and smFISH analysis indicates that differentiation occurs once immune cells depart the adipogenic niche. Sustained mitogenic signaling opposes adipocyte differentiation and thus niche exit could be required for differentiation to proceed (*Sun et al., 2017*). By the same token, we speculate that the failure of some proliferating ASC1 to differentiate might reflect persistent local interactions with resident or recruited immune cells.

The present findings highlight the complex cellular changes that occur in iBAT in response to cold exposure and support a new conceptual framework wherein brown adipocyte neogenesis is an adaptive response triggered when thermogenic demand exceeds thermogenic capacity (*Figure 10*). This framework implies mechanisms that sense metabolic stress within specific regions of BAT and translate those signals into adaptive hyperplasia and precise replacement of defective/exhausted BAs. While many of the mechanistic details remain to be uncovered, this physiological framework accounts for the facts that cold-induced neogenesis is delayed, transient, and localizes to specific locations in distinct tissue microenvironments. Consistent with this framework, available evidence indicates that the magnitude, location, and duration of neogenesis will vary directly with the discrepancy between thermogenic demand and capacity. For example, neogenesis induced by norepinephrine infusions in neurally intact iBAT is concentrated along the tissue border, whereas denervation-induced supersensitivity extends neogenesis into the tissue parenchyma (*Lee et al., 2015*). Compared to cold exposure, pharmacological activation of ADRB3 by CL appears to involve less of a shift in activation states of ASC1 and recruited immune cells (*Figure 5—figure supplement 1D-F* and data not shown), suggesting that such states might ultimately affect the differentiation fate of proliferating cells. Interestingly, we previously reported that CL treatment does not produce discernable neogenesis in 129S1/SvImJ mice (*Lee et al., 2012*), perhaps owing to greater induction of oxidative capacity in beige adipocytes (*Almind et al., 2007*) that might enhance thermogenic capacity without BA neogenesis.

In conclusion, our results indicate that cold exposure triggers the formation of dynamic adipogenic niches involving close, dynamic interactions between recruited myeloid cells and interstitial ASC1. These results demonstrate that neogenesis is not a reflexive response to ADRB1 activation in progenitors, as widely supposed, but rather is a coordinated adaptive response among BAs, immune cells, and progenitors that serves to match metabolic capacity to metabolic demand.

## Contact for reagents and resource sharing

Further information and requests for resources and reagents should be directed to and will be fulfilled by the Lead Contact, James Granneman (https://facultysenate.med.wayne.edu/profile/aa4929).

# Materials and methods

## Key resources table

| Reagent type (species) or resource | Designation | Source or reference | Identifiers | Additional information |
|---|---|---|---|---|
| Antibody | Anti-PDGFRalpha antibody (goat polyclonal) | R&D Systems | Mouse PDGFRalpha antibody; cat no. AF1062 | Diluted 1:70 in blocking buffer. |
| Antibody | Anti-GPNMB antibody (goat polyclonal) | R&D Systems | Mouse osteoactivin/GPNMB antibody; cat no. AF2330 | Dilute 1:400 in blocking buffer. |
| Antibody | Anti-MHCII antibody (rat monoclonal) | Biolegend | Alexa Fluor 647 rat anti-mouse I-A/I-E antibody; cat no. 107617 | Diluted 1:100 in blocking buffer. |
| Antibody | Anti-MKI67 antibody (rabbit monoclonal) | Invitrogen | Ki-67 recombinant rabbit monoclonal antibody; cat no. MAS-14520 | Diluted 1:125 in blocking buffer. |
| Antibody | Anti-PLIN1 antibody (goat polyclonal) | Everest Biotech | Goat anti-perilipin 1 (C terminus) antibody; cat no. EB07728 | Diluted 1:200 in blocking buffer. |
| Antibody | Anti-F4/80 antibody (rat monoclonal) | Bio-Rad | rat anti mouse F4/80 antibody; cat no. MCA497GA | Diluted 1:125 in blocking buffer. |
| Antibody | Anti-NNAT antibody (rabbit polyclonal) | Abcam | Anti-Neuronatin antibody; ab27266 | Diluted 1:400 in blocking buffer. |

*Continued on next page*

*Continued*

| Reagent type (species) or resource | Designation | Source or reference | Identifiers | Additional information |
|---|---|---|---|---|
| Antibody | Anti-IgG Alexa Fluor 647 (donkey polyclonal) | Abcam | Donkey anti-rat secondary antibody; cat no. ab150155 | Diluted 1:250 in blocking buffer. |
| Antibody | Anti-IgG Alexa Fluor 568 (donkey polyclonal) | Invitrogen | Donkey anti-goat secondary antibody; cat no. A11057 | Diluted 1:250 in blocking buffer. |
| Antibody | Anti-IgG Alexa Fluor 488 (donkey polyclonal) | Invitrogen | Donkey anti-goat secondary antibody; cat no. A11055 | Diluted 1:250 in blocking buffer. |
| Antibody | Anti-IgG Alexa Fluor 647 (donkey polyclonal) | Invitrogen | Donkey anti-goat secondary antibody; cat no. A21447 | Diluted 1:250 in blocking buffer. |
| Antibody | Anti-IgG Alexa Fluor 594 (donkey polyclonal) | Invitrogen | Donkey anti-rabbit secondary antibody; cat no. A21207 | Diluted 1:250 in blocking buffer. |
| Antibody | Anti-IgG Alexa Fluor 594 (goat polyclonal) | Invitrogen | Goat anti-rat secondary antibody; cat no. A11007 | Diluted 1:250 in blocking buffer. |
| Chemical compound, drug | EdU | Invitrogen | 5-Ethynyl-2'-deoxyuridine; cat no. A10044 | Diluted in sterile PBS. |
| Chemical compound, drug | Tamoxifen | Cayman Chemical | Tamoxifen; cat no. 13258 | Diluted in corn oil. |
| Chemical compound, drug | CL316,243 | Sigma-Aldrich | C5976 | |
| Chemical compound, drug | Paraformaldehyde (PFA) | Electron Microscopy Sciences | Paraformaldehyde 16% Solution, EM Grade; cat no. 15710 | Dilute to 4% PFA in PBS. |
| Chemical compound, drug | Sucrose | Fisher Scientific | D-Sucrose; cat no. BP220-1 | |
| Chemical compound, drug | O.C.T Compound | Fisher Heathcare | Tissue Plus O.C.T. Compound Clear; cat no. 23-730-572 | |
| Chemical compound, drug | Triton X-100 | Sigma | Triton X-100 laboratory grade; cat no. X100-500ML | |
| Chemical compound, drug | ProLong Gold antifade reagent | Invitrogen | ProLong Gold antifade reagent; cat no. P36930 | Coverslip mounting media |
| Chemical compound, drug | Collagenase Type 2 | Worthington Biochemical Corporation | Collagenase Type 2; cat no. LS004177 | Tissue digestion reagent |
| Chemical compound, drug | HBSS | Sigma-Aldrich | Hanks' Balanced Salt Solution; cat no. H1387−10 × 1 L | Tissue digestion reagent |
| Chemical compound, drug | HEPES | Sigma-Aldrich | cat no. H4034 | Tissue digestion reagent; FACs Buffer |
| Chemical compound, drug | Sodium Bicarbonate | Fisher Scientific | Sodium Bicarbonate (Powder/Certified ACS); cat no. S233-500 | Tissue digestion reagent; RBC lysis buffer |
| Chemical compound, drug | FF-BSA | Gemini Bio-products | Bovine Serum Albumin, Fatty Acid Free; cat no. 700−107 P | Tissue digestion reagent |
| Chemical compound, drug | EDTA | Millipore Sigma | Ethylenediaminetetraacetic acid (EDTA), Tetrasodium Tetrahydrate Salt; cat no. 13235-36-4 | Tissue digestion reagent; FACs Buffer |
| Chemical compound, drug | FBS | Atlanta Biologicals | Fetal Bovine Serum, Heat-inactivated; S11550H | FACs Buffer |
| Chemical compound, drug | Ammonium Chloride | Fisher Scientific | Ammonium Chloride (Crystalline/Certified ACS); cat no. A661-500 | RBC lysis buffer |
| Chemical compound, drug | TRIzol | Ambion by Life Technologies | TRIzol Reagent; cat no. 15596018 | |
| Chemical compound, drug | Dextran sulfate | Sigma-Aldrich | cat no. D8906 | |
| Commercial assay or kit | Click-iT EdU Cell Proliferation Kit for Imaging, Alexa Fluor 488 dye | Invitrogen | cat no. C10337 | |

*Continued on next page*

*Continued*

| Reagent type (species) or resource | Designation | Source or reference | Identifiers | Additional information |
|---|---|---|---|---|
| Commercial assay or kit | Debris Removal Solution | Miltenyi Biotec | cat no. 130-109-398 | |
| Commercial assay or kit | Lineage Cell Depletion Kit | Miltenyi Biotec | Lineage Cell Depletion Kit, mouse; cat no. 130-090-858 | |
| Commercial assay or kit | Chromium Single Cell 3' Library & Gel Bead Kit v2 | 10X Genomics | cat no. 120237 | |
| Commercial assay or kit | Chromium Single Cell 3' Library & Gel Bead Kit v3 | 10X Genomics | cat no. 1000092 | |
| Commercial assay or kit | Chromium Single Cell A Chip Kit | 10X Genomics | cat no. 120236 | |
| Commercial assay or kit | Chromium i7 Multiplex Kit | 10X Genomics | cat no. 120262 | |
| Commercial assay or kit | Bioanalyzer High Sensitivity DNA kit | Agilent Biotechnologies | 2,100 Bioanalyzer High Sensitivity DNA kit; cat no. 5067–4626 | |
| Commercial assay or kit | Direct-zol RNA Miniprep Kit | Zymo Research | cat no. R2050 | |
| Commercial assay or kit | High Sensitivity D1000 ScreenTape for DNA Analysis | Agilent Biotechnologies | cat no. NC1786959 | |
| Commercial assay or kit | QuantSeq 3' mRNA-Seq Library Prep Kit FWD for Illumina | Lexogen | cat no. 113.96 | |
| Commercial assay or kit | KAPA SYBR Fast Universal qPCR Kit – Illumina Kapa | Kapa Biosystems | cat no. 07960441001 | |
| Commercial assay or kit | High Capacity cDNA Reverse Transcription Kit | Applied biosystems | cat no. 4368814 | |
| Commercial assay or kit | DyNAmo HS SYBR Green qPCR Kit | Thermoscientific | cat no. F-410XL | |
| Commercial assay or kit | TrueVIEW | Vector Laboratories | cat no. SP-8400 | |
| Other | Micro-osmotic Pump | ALZET | Micro-osmotic Pump, 0.5 uL/hr; cat no. 1007D | Experimental model and subject details; continuous CL infusion |
| Other | Normal Donkey Serum | Fisher | Jackson Immunoresearch SERUM NORMAL DONKEY; cat no. 017-000-121 | Tissue processing and immunohistochemistry; blocking buffer |
| Other | Normal Goat Serum | Sigma | cat no. G-9023 | Tissue processing and immunohistochemistry; blocking buffer |
| Other | Lens Culinaris Agglutinin (LCA) DyLight 649 | Vector Laboratories | DL-1048 | Diluted 1:150 in blocking buffer. |
| Other | HCS LipidTOX Deep Red neutral lipid stain | Invitrogen | cat no. H34477 | Diluted 1:500. |
| Other | DAPI | Sigma | D9542 | 20 mg/mL; dilute 1:2,000 |
| Other | 100 μm Sterile Cell Strainer | fisherbrand | Sterile Cell Strainer 100 μm; cat no. 22-363-549 | Isolation of stromal vascular cells for mouse iBAT single cell |
| Other | 40 μm Sterile Cell Strainer | fisherbrand | Sterile Cell Strainer 40 μm; cat no. 22-363-547 | Isolation of stromal vascular cells for mouse iBAT single cell |
| Other | MS Columns | Miltenyi Biotec | cat no. 130-042-201 | Magnetic bead enrichment of cellular subtypes |
| Other | OctoMACS Separator | Miltenyi Biotec | cat no. 130-042-109 | Magnetic bead enrichment of cellular subtypes |
| Other | Chromium Controller Instrument | 10X Genomics | cat no. 1000204 | Single-cell RNA library preparation and sequencing |
| Other | Bst LF polymerase | New England Biolabs | cat no. M0275L | Single-molecule fluorescence in situ hybridization (smFISH) |

*Continued on next page*

*Continued*

| Reagent type (species) or resource | Designation | Source or reference | Identifiers | Additional information |
|---|---|---|---|---|
| Other | Digest-All 3 | ThermoFisher | cat no. 003009 | Single-molecule fluorescence in situ hybridization (smFISH) |
| Software, algorithm | Cell Ranger v3.0.1 | 10X Genomics | | |
| Software, algorithm | SoupX v1.0.1 | PMID:33367645 | | |
| Software, algorithm | Seurat v3.1.5 | PMID:31178118 | | |
| Software, algorithm | CellPhoneDB v2.1.3 | PMID:32103204 | | |
| Software, algorithm | NicheNet v0.1.0 | PMID:31819264 | | |
| Software, algorithm | CASAVA v1.8.2 | Illumina | | |
| Software, algorithm | STAR-2.6.1d | PMID:23104886 | | |
| Software, algorithm | Htseq v0.11.2 | PMID:25260700 | | |
| Software, algorithm | iDEP v0.95 | PMID:30567491 | | |
| Software, algorithm | GraphPad Prism v9.3.1 | GraphPad Software | | |
| Software, algorithm | Imaris Viewer x64 v9.9.0 | Oxford Instruments Group | | |
| Strain, strain background (*Mus musculus*) | C57 | Jackson Labs | C57BL/6J; stock no. 000664 | |
| Strain, strain background (*Mus musculus*) | LSL-tdTomato | Jackson Labs | B6.Cg-Gt(ROSA)26Sor_tm9(CAG-tdTomato)Hze/J; stock no. 007909 | |
| Strain, strain background (*Mus musculus*) | Pdgfra-CreER$^{T2}$ | Jackson Labs | B6.129S-Pdgfratm1.1(cre/ER$^{T2}$)Blh/J; stock no. 032770 | |
| Strain, strain background (*Mus musculus*) | *Adrb1$^{fl/fl}$* | PMID:27548523 | | Dr. Jeffery Zigman, The University of Texas Southwestern Medical Center |

## Experimental model and subject details

C57BL/6J (C57; stock no. 000664), B6.129S-*Pdgfra$^{tm1.1(cre/ERT2)Blh}$*/J (Pdgfra-CreER$^{T2}$ mice; stock no. 032770), and B6.Cg-Gt(ROSA)26Sor_tm9(CAG-tdTomato)Hze/J (LSL-tdTomato; stock no. 007909) mice were purchased from the Jackson Laboratory. Floxed Adrb1 mice (*Adrb1$^{fl/fl}$*) (*Mani et al., 2016*) were kindly provided by Dr. Jeffrey Zigman (The University of Texas Southwestern Medical Center). All mice were housed at 24 °C +/- 2 °C with a 12:12 light-dark cycle in an AAALAC approved animal facility at Wayne State University (Detroit, MI). Mice were fed a standard chow diet ad libitum (LabDiet 5L0D, PMI Nutrition International, Brentwood, MO). Animal protocols were approved by the Institutional Animal Care and Use Committee at Wayne State University (#16-03-055 and #19-03-1024). All mice were euthanized at 8–14 weeks of age. Animals were randomly assigned to treatment groups. For cold exposure experiments, mice were housed in a rodent incubator (Powers Scientific, Inc) set to 6 °C for up to 5 days or maintained at colony room temperature as controls. In the incubators, mice were housed individually with no nesting materials in static cages. Mice were euthanized by $CO_2$ asphyxiation and cervical dislocation.

For continuous ADRB3 stimulation, mice were subcutaneously implanted with micro-osmotic pumps (ALZET) containing the ADRB3 agonist CL316,243 (Sigma-Aldrich) which infused the animals at a rate of 0.75 nmol/hour for four days.

For 5-ethynyl-2'-deoxyuridine (EdU) labeling of proliferating cells, mice were injected with EdU (Invitrogen, 20 mg/kg, i.p.) prepared in sterile PBS. For flash labeling of proliferating ASCs, EdU was administered once on the third day of cold exposure, and animals were euthanized two hours later. For tracing differentiated adipocytes, EdU was administered on either day three of cold exposure, or on days three and four of cold exposure, and animals were euthanized on day five. Number of EdU injections is indicated in the figure.

Cre recombination in Pdgfra-CreER$^{T2}$ x LSL-tdTomato mice was induced by administering tamoxifen dissolved in corn oil (Cayman Chemical, 100 mg/kg, oral gavage) once per day for 3–4 consecutive

days. Cold exposure studies were started 1 week after the last dose of tamoxifen. To knockout ADRB1 in PDGFRA+ ASCs in vivo, *Adrb1*<sup>fl/fl</sup> mice were crossed with Pdgfra-CreER<sup>T2</sup> mice to create mice homozygous for floxed *Adrb1* and heterozygous for Pdgfra-CreER<sup>T2</sup> (*Adrb1*<sup>fl/fl</sup> x Pdgfra-CreER<sup>T2 +/-</sup> mice (*Adrb1* KO)). *Adrb1* was knocked out in PDGFRA+ cells by induction of Pdgfra-CreER<sup>T2</sup> by tamoxifen dissolved in sunflower oil (Cayman Chemical, 100 mg/kg, oral gavage) once per day for 5 consecutive days. Littermates without a Pdgfra-CreER<sup>T2</sup> allele (*Adrb1*<sup>fl/fl</sup> x Pdgfra-CreER<sup>T2 -/-</sup> mice) were used as controls and also administered tamoxifen. Cold exposure studies were started 2–3 weeks after the last dose of tamoxifen.

## Tissue processing and immunohistochemistry

Tissues were fixed with 4% paraformaldehyde overnight at 4 °C, transferred through a sucrose gradient, embedded in O.C.T. compound, and cut into 20 µm-thick sections. Immunostaining was performed on fixed frozen tissue. Samples were pre-incubated with permeabilization buffer (0.3% TritonX 100 in PBS) for 30 minutes at RT and blocking buffer (5% serum corresponding to the species of the secondary antibody with 0.1% TritonX 100 in PBS) for 1 hour at RT, and then incubated with primary antibody diluted in blocking buffer overnight. Primary antibodies and dilutions used for immunohistochemistry in this study were the following: PDGFRA (goat, 1:70; R&D Systems); GPNMB (goat, 1:400; R&D Systems); I-A/I-E Alexa Fluor 647 (MHCII) (rat, 1:100; Biolegend); MKI67 (rabbit, 1:125; Invitrogen); perilipin1 (PLIN1) (goat, 1:200; Everest Biotech); F4/80 (rat, 1:125; Bio-Rad); lens culinaris agglutinin, DyLight 649 (LCA) (1:150, Vector Laboratories); NNAT (rabbit, 1:400, abcam). After three washes, slides were incubated with the following secondary antibodies, diluted in blocking buffer for one hour at RT: donkey anti-rat Alexa Fluor 647 (1:250; abcam); donkey anti-goat Alexa Fluor 568 (1: 250; Invitrogen); donkey anti-goat Alexa Fluor 488 (1: 250, Invitrogen); donkey anti-goat Alexa Fluor 647 (1: 250, Invitrogen); donkey anti-rabbit Alexa Fluor 594 (1: 250, Invitrogen); goat anti-rat Alexa Fluor 594 (1: 250, Invitrogen). HCS LipidTOX Deep Red Neutral Lipid Stain (1:500; Invitrogen) was used for lipid staining and added with secondary antibodies. For EdU detection, after antibody staining, slides were stained using the Click-iT EdU Imaging Kit (Invitrogen) following the manufacturer's instructions. Samples were incubated with Click-iT reaction cocktail for 30 min, covered. After washing, samples were counterstained with DAPI (1:5,000; Sigma). Slides were coverslipped in ProLong Gold antifade reagent (Invitrogen) and examined by fluorescence microscopy. Staining where primary antibodies were omitted was used as nonspecific controls for immunohistochemistry.

## Single-molecule fluorescence in situ hybridization (smFISH)

smFISH was conducted on fixed-frozen iBAT samples following the recently published SABER-FISH protocol (*Kishi et al., 2019*), with some modifications. Briefly, gene-specific probe sets, branch probes, the Clean.G oligo, and the Primer Exchange Reaction (PER) hairpin oligos were purchased from Invitrogen, and fluorescent-conjugated oligos were purchased from Integrated DNA Technologies or Sigma-Aldrich (St. Louis, MO). Primary probes for *Dcn*, *Top2a*, *Nnat*, *Bmper*, *Pi16*, *Gdf10*, *H2-Ab1* (MHCII), *Adrb1*, and *Gpnmb* were designed using the mm10 reference genome as described by *Kishi et al., 2019*. Primer sequences are presented in *Supplementary file 8*. Primary probes were concatenated to a length of ~500–750 nt and branch probes to ~500 nt in a PCR reaction containing Bst LF polymerase (640 U/mL; New England Biosciences), 300 µM each of dNTPs (dATP, dCTP, dTTP), and 10 mM MgSO4, as described previously by *Kishi et al., 2019* in the Supplementary Protocols (*Kishi et al., 2019*). After synthesis, all probes were purified using MinElute PCR purification columns (Qiagen) prior to use.

iBAT tissue was excised and fixed in 10% neutral buffered formalin (NBF) for 5–7 hr at room temperature, then transferred to 30% sucrose in PBS (pH 7.4) overnight at 4 °C (12–16 hr). Cryosections (20 µm) were affixed onto Superfrost plus charged slides (Fisher Scientific) and sections preprocessed following the RNAscope technical note (TN 320534) up through the heat (antigen retrieval) step. The antigen retrieval buffer used was 10 mM citrate (pH 6.0). Sections were then lightly digested with pepsin (Digest-All 3, ThemoFisher) for 7 minutes at 40 °C, rinsed in PBS (pH 7.4), and preincubated in hybridization wash buffer (40% formamide, 2X SSC (pH 7), and 1% Tween-20) for 3–5 hr at 42 °C. Primary probes were added at 1 µg final concentration and hybridized at 42 °C for at least 16 hr. The following day, sections were washed in hybridization wash buffer and then incubated in branch hybridization solution (25% formamide, 2X SSC, 1% Tween-20, and 10% dextran sulfate) containing

branch probes (100 nM) for 5 hr at 37 °C. After washing in hybridization wash buffer (25% formamide), sections were incubated with fluorescent oligos (~500 nM each) for 2 hr at 37 °C. Sections were then washed, stained with DAPI, and treated with TrueView (Vector labs) prior to mounting.

## Microscopy and image analysis

Immunofluorescence microscopy was performed either using (1) an Olympus IX-81 microscope equipped with a spinning disc confocal unit and 10X, 20X, 40X (0.9NA) water immersion, and 60X (1.2NA) water immersion objectives, using standard excitation and emission filters (Semrock) for visualizing DAPI, FITC (Alexa Fluor 488), Cy3 (Alexa Fluor 568, 594), and Cy5 (Alexa Fluor 647, Dy-Light 649, LipidTOX), (2) a Keyence microscope (BZ-X810; Keyence; Itasca, IL) with a 40X (0.6NA) air objective, or (3) an Andor Dragonfly spinning disk confocal microscope (Andor; Belfast, UK) using 40X (1.1NA) water and 63X (1.4NA) oil objectives. Where stated, tissue autofluorescence was captured using a cyan fluorescent protein filter set (Chroma 31044V2). Raw data of single optical sections or confocal Z-stacks were processed using cellSens imaging software (Olympus). Quantification of the distance between immune cell types (GPNMB+ MAC1 and MHCII+ dendritic cells) and tdTomato+ EdU+ ASCs/adipocytes was performed on 20X single optical sections using the Measurement and ROI toolbar Arbitrary Line function in cellSens. smFISH images acquired with the Dragonfly microscope were processed using Fusion software (v 2.3.0.44; Andor), and then analysis and quantification performed with the Imaris Viewer x64 (v9.9.0) software using 3D view. For the *Bmper/Pi16/Nnat* analysis (*Figure 4—figure supplement 1B*), images of 20 μm-thick sections from three individual cold-exposed mice were analyzed, totaling at least 90 *Nnat+* cells per animal. For each *Nnat+* cell, co-expression of *Bmper* or *Pi16* were recorded. For the *Adrb1* molecule analysis (*Figure 5—figure supplement 1C*), images from two individual cold-exposed mice were analyzed per genotype (WT or *Adrb1* KO). *Nnat+* cells were identified, and the number of *Adrb1* molecules (puncta) in the same focal plane as *Nnat* expression was documented. For the neighbor analysis (*Figure 8*), images from three individual cold-exposed mice were analyzed, totaling at least 100 cells for analysis. Cells within a 20 μm diameter circle centered around the cell type of interest (*H2-Ab1+*, *Nnat+*, *Dcn+*, or *Dcn+ Top2a+*) were documented. Analysis was performed by excluding classified cells with no neighbors within 20 μm.

## Single-cell RNA sequencing experiments

### Isolation of stromal vascular cells for mouse iBAT single-cell analysis

For cold exposure and CL treatment experiments, male C57 mice were exposed to cold (6 °C) or infused with CL for four days, or maintained as room temperature controls. For *Adrb1* KO experiments, female *Adrb1*^fl/fl x Pdgfra-CreER^T2+/- mice (*Adrb1* KO) and *Adrb1*^fl/fl x Pdgfra-CreER^T2-/- genotype controls, all treated with tamoxifen, were exposed to cold for four days or maintained at room temperature. Interscapular brown adipose tissues were surgically removed and processed for stromal vascular cell (SVC) isolation. Tissues of 3–4 mice were pooled after digestion for SVC isolation, similar to methods previously described (*Burl et al., 2018*; *Lee et al., 2012*; *Figure 2—figure supplement 1A*). Briefly, following dissection, iBAT pads were washed with PBS, minced, and digested with 2 mg/mL type 2 collagenase (Worthington Biochemical Co.) in Hanks' balanced salt solution (HBSS; Sigma-Aldrich) containing 4 mM sodium bicarbonate, 10 mM HEPES (pH 7.4, Gibco; Sigma-Aldrich) and 0.5% fatty acid free bovine serum albumin (FF-BSA; Gemini Bioproducts, West Sacramento, CA) for 30 min at 37 °C (*Figure 2—figure supplement 1A*). Ethylenediaminetetraacetic acid (EDTA) was added to a final concentration of 10 mM and samples were incubated for an additional 5 min to promote complete dissociation of SVCs.

Digested cell samples from the same treatment group were combined and filtered through 100 μm and 40 μm sterile cell strainers. The samples were washed with PBS buffer containing 1 mM EDTA, 2.5 mM HEPES, and 10% heat-inactivated fetal bovine serum (FBS, Atlanta Biologicals Inc) (FACS buffer) and centrifuged at 500 x *g* for 10 min at 4 °C. After removing the supernatant, pelleted stromal vascular cells were incubated in red blood cell lysis buffer containing 14 mM sodium bicarbonate, 0.154 M ammonium chloride, and 0.1 mM EDTA for 5 min at room temperature, then passed through a 100 μm sterile cell strainer. Cells were collected by centrifugation at 500 x *g* for 10 min at 4 °C. Cellular debris was removed by density gradient centrifugation using Debris Removal Solution (Miltenyi Biotec). Cells recovered were resuspended in PBS containing 0.5% FF-BSA.

## Magnetic bead enrichment of cellular subtypes

iBAT SVCs were separated into lineage marker positive (Lin+) and lineage-marker negative (Lin-) cell fractions using magnetic bead cell separation (MACS) (*Figure 2—figure supplement 1A*). SVCs were labeled using the mouse Lineage Cell Depletion kit (Miltenyi Biotec) according to the manufacturer's instructions. This kit contains anti-CD5, anti-CD11b, anti-CD45R (B220), anti-Gr-1 (Ly-6G/C), anti-7–4, and anti-Ter-119 antibodies. Labeled samples were passed onto MS columns (Miltneyi Biotec) on the OctoMACS separator (Miltenyi Biotec); the flow through was collected as the Lin- cell fraction and bound Lin+ cells were eluted from the column. Samples were centrifuged at 500 x *g* for 10 min at 4 °C and pelleted cells were resuspended in PBS with 0.04% FF-BSA. Cells were counted and diluted to a concentration of 1,000 cells/uL.

## Single-cell RNA library preparation and sequencing

Single-cell libraries for all samples were prepared using the 10X Genomics Chromium Single Cell 3' Reagent Kit v2, apart from *Adrb1* KO and WT RT libraries which were prepared with Reagent Kit v3 (*Figure 2—figure supplement 1C*). Following MACS, single-cell suspensions were loaded onto the Single Cell Chip A. An estimated 10,000 cells were loaded per lane for an expected recovery of ~6,000 cells per library. The assembled chip was placed on a 10X Genomics Chromium Controller Instrument (10 X Genomics) to generate single-cell gel beads in emulsion (GEMs). Single-cell RNA-seq libraries were prepared according to the manufacturer's instructions. Libraries were quantified using the Kappa PCR kit (Kappa Biosystems) and sequenced with the Illumina NextSeq500 using high output 75-cycle kits with the following read length configuration: v2 libraries 26 bp read1, 8 bp I7 index, and 58 bp read2; v3 libraries 28 bp read1, 8 bp I7 index, and 56 bp read2.

10X Genomics Cell Ranger (v3.0.1) was used to perform sample demultiplexing, alignment, filtering, and UMI counting (*Figure 2—figure supplement 1B*). Count files were processed using SoupX (v1.0.1) (*Young and Behjati, 2020*), an R package for the estimation and removal of cell-free mRNA contamination in single-cell data (*Figure 2—figure supplement 1B*). For SoupX library cleanup, *Fabp4* and *Car3* were used as the non-expressed genes, as we expect the largest contribution of contamination to come from damaged mature BAs that have been excluded from these libraries, and clusters = FALSE. Corrected count libraries were input into the R program Seurat (v3.1.5) (*Stuart et al., 2019*) and libraries were quality filtered (Lin+: percent.mt <10 & nCount_RNA < 15000 & nFeature_RNA < 4000; Lin-: percent.mt <10 & nCount_RNA < 15000 & nFeature_RNA > 100) (*Figure 2—figure supplement 1B*). For the C57 RT, COLD, and CL libraries, after normalization, libraries from the same replicate experiment were merged (*i.e.* Replicate1_Lin-_RT + Replicate1_Lin-_COLD = Replicate1_merge). The top 2,000 variable genes were identified for each merged object, and objects from similar cell fractions (Lin+ or Lin-) were integrated to align common features of the dataset. For the *Adrb1* WT and KO libraries, because the libraries for different treatments were collected on different days and with different version of the 10X kit, we integrated all of the libraries in Seurat to correct for these effects and perform comparative scRNA-seq across the libraries. Therefore, after normalization we identified the top 2,000 variable genes for each library and all libraries were integrated. For all integrated objects, we performed linear dimensional reduction, cell clustering, and data visualization using t-distributed stochastic neighbor embedding (t-SNE). Differentially expressed genes that define each cluster were identified using a Wilcoxon Rank Sum test in Seurat with the following parameters: min.pct=0.20, logfc.threshold=0.2, only.pos=TRUE (*Figure 2—figure supplement 1B*). These markers were used to assign cell type identity. Gene Ontology (GO) analysis of DEGs and p-values were produced by the *Gene Ontology Consortium, 2021*; *Ashburner et al., 2000*. Volcano plots were prepared from DEGs calculated by distinguishing ASCs from control libraries with ASCs from cold libraries (*i.e.* ASC2_controls vs. ASC2_cold). Colored points and quantification displayed on the plot are the number of genes with an adjusted p-value less than 0.05 and the absolute value of the fold change greater than 0.5.

## Ligand-Receptor Analysis

To identify potential ligand-receptor pairs in our single-cell dataset, we used the computational programs CellPhoneDB (v2.1.3) (*Efremova et al., 2020*) and NicheNet (v0.1.0) (*Browaeys et al., 2020*). CellPhoneDB is a curated repository of ligands and receptors that interrogates single-cell data for expression of these pairs. The NicheNet repository utilizes a model that incorporates intracellular

signaling to prioritize pairs. In addition to these computational programs, we generated lists of genes expressed in each cell type and cross-referenced them with two databases: the NicheNet ligand-receptor database and the mouse ligand and receptor database compiled in *Skelly et al., 2018*. The results from each method were compiled into a master list of potential ligand-receptor pairs. Finally, every ligand and receptor were manually plotted in our single-cell dataset to remove those with expression not specific to the MAC/DEND cells (ligands) and the ASC1_cold-activated cells (receptors).

## Cold exposure time course for whole tissue RNA isolation and sequencing

RNA was isolated from mouse iBAT following the standard protocol for the Direct-zol RNA Miniprep kit (Zymo Research) and assessed for quality on the Agilent 2200 TapeStation (Agilent Technologies). Transcriptome profiles were generated using the QuantSeq 3' mRNA-Seq Library Prep Kit FWD for Illumina (Lexogen) with 250 ng of total RNA. Libraries were sequenced on an Illumina NovaSeq 6000 using a SP 100 cycle NovaSeq flow cell with read length configuration 76 bp with dual indexing: 12 bp for i7 and i5 indexing. Sequencing data was demultiplexed using Illumina's CASAVA 1.8.2 software.

Sequencing reads were aligned to the mouse genome with STAR-2.6.1d (*Dobin et al., 2013*). HTseq (v0.11.2) (*Anders et al., 2015*) was used to determine the read counts per gene based on Ensembl gene annotations from mouse Gencode release 38 (GRCm38.98). We sequenced a total of ~171.3 million reads, with an average of ~4.9 million reads per library. Raw count data was uploaded into iDEP (*Ge et al., 2018*), a web application for analysis of RNA-seq data. Data was transformed in iDEP (v0.95) using the rlog function. K-means clustering was performed on the top 2,000 DEGs with k=4 clusters. Analysis focused on the clusters of genes upregulated with cold exposure.

## Gene expression analysis in iBAT of *Adrb1* KO mice

Whole-tissue RNA was extracted from the iBAT of control (*Adrb1*$^{fl/fl}$ (WT)) or *Adrb1* KO mice after being maintained at room temperature or exposed to cold for four days using TRIzol (Invitrogen). mRNA was reverse-transcribed using the High-Capacity cDNA Reverse Transcription Kit (ThermoFisher Scientific). Fifty nanograms of cDNA was analyzed in a 20 uL quantitative PCR reaction (DyNAmo HS SYBR Green qPCR Kit, ThermoFisher Scientific) with 500 nM of primers. Quantitative PCR was performed using the AriaMx Real-Time PCR System (Agilent Technologies). Expression data was normalized to the housekeeping gene ribosomal protein lateral stalk subunit P0 (*Rplp0*). *Rplp0* was amplified using the primers 5'- AGATTCGGGATATGCTGTTGGC-3' (forward) and 5'- TCGGGTCCTAGACCAGTGTTC-3' (reverse). *Top2a* was amplified using primers 5'-GATGGTTTTACGGAGCCAGTTTT-3' (forward) and 5'-CACGTCAGAGGTTGAGCACT-3' (reverse). *Gpnmb* was amplified using primers 5'-CTATCCCTGGCAAAGACCCA-3' (forward) and 5'-GGCTTGTACGCCTTGTGTTT-3' (reverse). *Trem2* was amplified using primers 5'-AGCACCTCCAGGCAGGTTT-3' (forward) and 5'-TTGATTCCTTGGAAAGAGGAGGA-3' (reverse). *Birc5* was amplified using primers 5'-ACCGAGAACGAGCCTGATTT-3' (forward) and 5'-ATGCTCCTCTATCGGGTTGTC-3' (reverse).

## Statistical analysis

Statistical analyses were performed using GraphPad Prism v9.3.1 software (GraphPad Software, San Diego, CA, USA), R (*v 4.0.3*) ("*R Development Core Team, 2020*") and RStudio (*R Studio Team, 2022*), or an online Chi-square Calculator (*Stangroom, 2002*). Data are presented as mean ± SD or mean ± SEM as indicated in the figure legends. Comparison among groups was performed using a 2-way ANOVA on log transformed data. Pearson correlation coefficient was calculated in R using cor() function and reported p-values were calculated using the cor.test() function.

## Acknowledgements

The authors thank the Luca and Pique-Regi lab at Wayne State University, especially Adnan Alazizi, for their assistance with sequencing. Additionally, we thank the Genome Sciences Core at Wayne State University, especially Dr. Katherine Gurdziel, for their assistance with the whole tissue RNA-sequencing experiments. Lastly, we are greatly appreciative of Dr. Carey Lumeng (University of Michigan) for providing his expertise on adipose tissue immune cells and members of CIMER (Wayne State University) for their useful suggestions.

# Additional information

### Funding

| Funder | Grant reference number | Author |
|---|---|---|
| National Institute of Diabetes and Digestive and Kidney Diseases | R01-DK062292 | James G Granneman |
| National Institute of Diabetes and Digestive and Kidney Diseases | F31-DK116536 | Rayanne B Burl |

The funders had no role in study design, data collection and interpretation, or the decision to submit the work for publication.

### Author contributions

Rayanne B Burl, Data curation, Formal analysis, Funding acquisition, Validation, Investigation, Visualization, Methodology, Writing – original draft, Writing – review and editing; Elizabeth Ann Rondini, Data curation, Formal analysis, Validation, Investigation, Visualization, Methodology, Writing – original draft, Writing – review and editing; Hongguang Wei, Validation, Investigation, Methodology, Writing – review and editing; Roger Pique-Regi, Supervision, Methodology, Writing – review and editing; James G Granneman, Conceptualization, Resources, Formal analysis, Supervision, Funding acquisition, Validation, Investigation, Visualization, Methodology, Writing – original draft, Writing – review and editing

### Author ORCIDs

Rayanne B Burl (ID) http://orcid.org/0000-0002-3905-4095
Roger Pique-Regi (ID) http://orcid.org/0000-0002-1262-2275
James G Granneman (ID) http://orcid.org/0000-0001-7013-6630

### Ethics

All animal protocols were approved and conducted in accordance with the Institutional Animal Care and Use Committee at Wayne State University (#16-03-055 and #19-03-1024).

### Decision letter and Author response

Decision letter https://doi.org/10.7554/eLife.80167.sa1
Author response https://doi.org/10.7554/eLife.80167.sa2

# Additional files

### Supplementary files

• Supplementary file 1. DEGs for each cluster in the CONTROL and COLD Lin- iBAT data.

• Supplementary file 2. DEGs for each cluster in the reclustered CONTROL and COLD ASC iBAT data.

• Supplementary file 3. DEGs for each cluster in the WT and *Adrb1* KO Lin- iBAT data.

• Supplementary file 4. DEGs for each cluster in the reclustered WT and *Adrb1* KO ASC iBAT data.

• Supplementary file 5. DEGs for each cluster in the CONTROL, COLD, and CL Lin- iBAT data.

• Supplementary file 6. DEGs for each cluster in the reclustered CONTROL, COLD, and CL ASC iBAT data.

• Supplementary file 7. DEGs for each cluster in the CONTROL and COLD Lin+ iBAT data.

• Supplementary file 8. List of gene-specific primers used to synthesize probes for SABER-FISH.

• MDAR checklist

### Data availability

Sequencing data from this study has been deposited in the Gene Expression Omnibus (GEO) under the series accession number GSE207707. Scripts for data processing will be made available through GitHub (https://github.com/RBBurl1227/eLife-2022-ColdInducedBrownAdipocyteNeogenesis, (copy archived at swh:1:rev:eb5322ae1f67ed1f23c59210ed9ed5446566fd22)).

The following dataset was generated:

| Author(s) | Year | Dataset title | Dataset URL | Database and Identifier |
|---|---|---|---|---|
| Granneman JG | 2022 | Deconstructing cold-induced brown adipocyte neogenesis in mice | http://www.ncbi.nlm.nih.gov/geo/query/acc.cgi?acc=GSE207707 | NCBI Gene Expression Omnibus, GSE207707 |

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

## Appendix 1

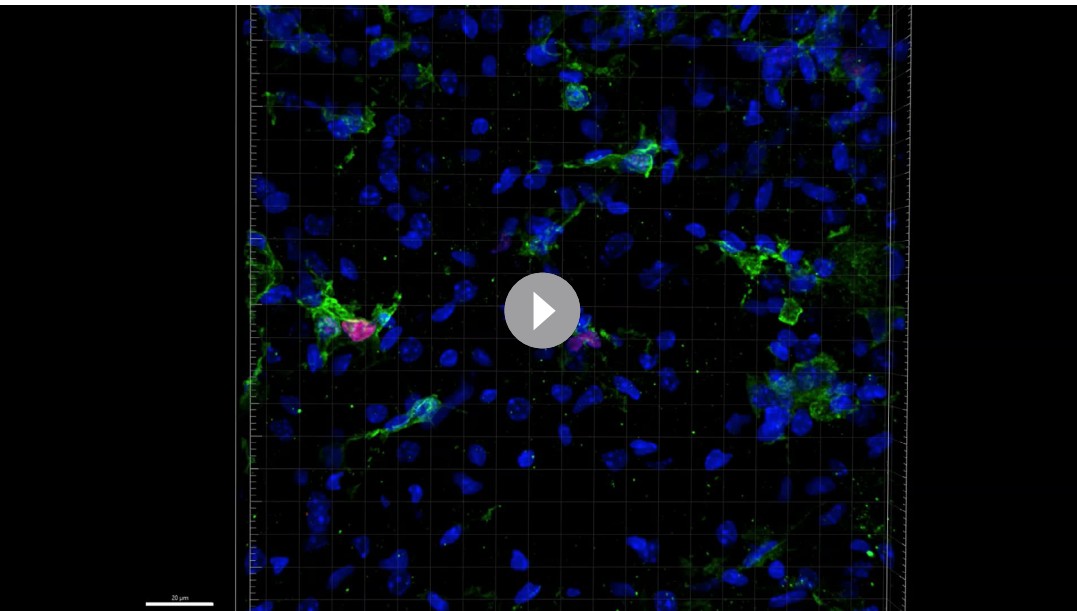

**Appendix 1—video 1.** Video showing 3D detail of *Figure 7B*. High-magnification movie of iBAT fixed-frozen tissue section from cold-exposed mice stained for PDGFRA (green), F4/80 (purple), and MKI67 (red). Nuclei were counterstained with DAPI. Video shows sequential addition of the following elements: (1) distribution of cell nuclei stained with DAPI, (2) proliferating (MKI67, red) PDGFRA+ progenitors (green), (3) F4/80+ dendritic cells (pink). Note that the cellular processes of each proliferating PDGFRA+ progenitor (yellow arrow heads in *Figure 7B*) extends toward, and makes apparent contact with F4/80 dendritic cells.

https://elifesciences.org/articles/80167/figures#video1

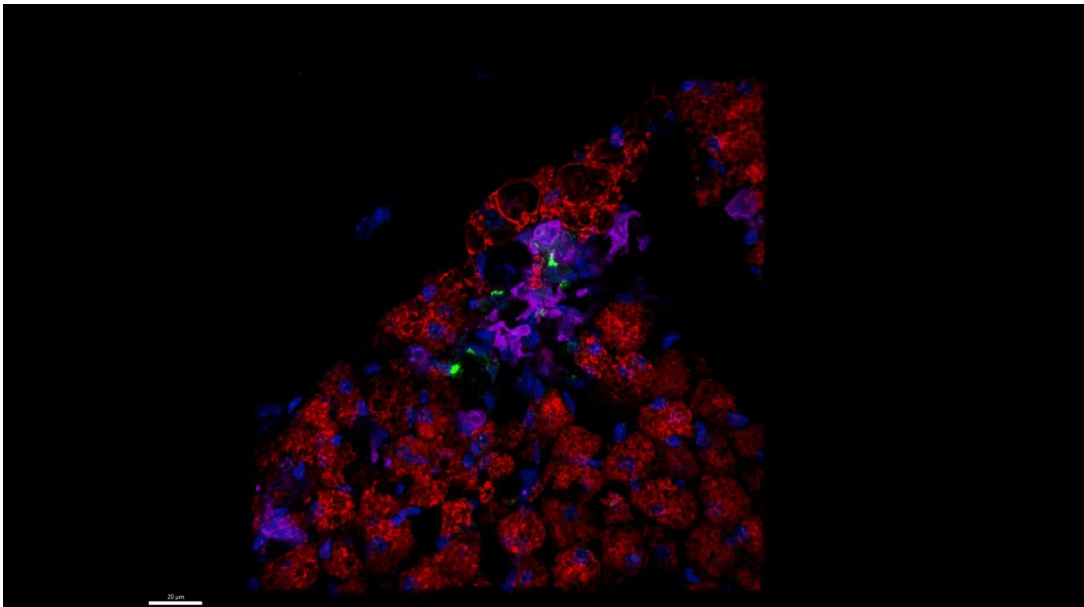

**Appendix 1—video 2.** High-magnification movie of iBAT fixed-frozen tissue section from cold-exposed mice stained for GPNMB (green), MHCII (purple), and PLIN1 (red). Nuclei were counterstained with DAPI. Video shows sequential addition of the following elements: (1) image with immune marker channels turned off to show the outline of the effercytosis site (ES). (2) GPNMB+ cells (green) in the ES. (3) MHCII+ cells (purple) in the ES. Note that both GPNMB+ cells and MHCII+ cells surround cellular vacancies.

https://elifesciences.org/articles/80167/figures#video2

