## [Editor Report]

This study elucidates transcriptional profiles of the stromal vascular fraction of murine brown adipose tissue in the context of thermogenic stimulation. The authors combine systems and reductionist approaches to show the reliance of mature brown adipocytes on adrenergic activation to indirectly stimulate progenitor proliferation and differentiation. This timely work will provide beneficial data for public use and further resolve the complexities underlying brown adipose physiology.

---

## [Decision Letter]

**Decision letter after peer review:**

Thank you for submitting your article "Deconstructing cold-induced brown adipocyte neogenesis in mice" for consideration by *eLife*. Your article has been reviewed by 3 peer reviewers, including Peter Tontonoz as Reviewing Editor and Reviewer #1, and the evaluation has been overseen by Carlos Isales as the Senior Editor. The following individual involved in review of your submission has agreed to reveal their identity: Patrick Seale (Reviewer #3).

Essential revisions:

The reviewers do not believe that any additional experiments are required. Please respond to the discussion points and technical concerns in a revised manuscript.

*Reviewer #1 (Recommendations for the authors):*

A pseudotime or latent time analysis (RNA velocity) would be an effective way to present the data showing ASC1 commitment to brown adipogenesis. CellRank is a newer method which can integrate time course, RNA velocity, and clustering analyses to determine terminal states of cells that can be easily adapted to the authors existing Seurat pipeline, if they so choose. Usage of a splicing-aware aligner can generate gene expression matrices of spliced and unspliced transcripts that can be appended to existing Seurat objects as assays and exported into Python for analysis with scVelo [PMID: 32747759] and CellRank [PMID: 35027767].

*Reviewer #2 (Recommendations for the authors):*

1. Please explain why mice were adapted to RT rather than thermoneutrality for the control mice used in the single cell experiments (since RT mice are also cold)? Could this affect interpretations?

2. It would be nice to include functional data showing that the ASC1 population is the exclusive source of brown adipocytes.

3. It was surprising that the authors did not test if KO of Adrb1 in other populations (e.g. BAs) inhibits NE-stimulated neogenesis as they showed by global KO that it is required. What's the key cell population?

4. These data are beautiful and establish interesting hypotheses, though I would have liked to see more functional analysis of potential mechanisms, for example between immune-BA interactions. Nevertheless, I do think this is appropriate for *eLife*.

*Reviewer #3 (Recommendations for the authors):*

Overall, I find the paper to be very compelling and strongly support publication in *eLife* without delay.

---

## [Author Response]

Reviewer #1 (Recommendations for the authors):A pseudotime or latent time analysis (RNA velocity) would be an effective way to present the data showing ASC1 commitment to brown adipogenesis. CellRank is a newer method which can integrate time course, RNA velocity, and clustering analyses to determine terminal states of cells that can be easily adapted to the authors existing Seurat pipeline, if they so choose. Usage of a splicing-aware aligner can generate gene expression matrices of spliced and unspliced transcripts that can be appended to existing Seurat objects as assays and exported into Python for analysis with scVelo [PMID: 32747759] and CellRank [PMID: 35027767].

We thank the reviewer for recommendations regarding pseudotime/latent analysis and recognize the recent popularity of these analyses for presentation of scRNA-seq data. We are aware of, and have applied, several pseudotime programs (Slingshot, scVelo, Monocle) to our datasets (not shown); however, we have found that predictions among different programs are sometimes contradictory, and findings are not always supported by experimental data. Inconsistencies and limitations in model estimations and assumptions have recently been raised by others (https://doi.org/10.1101/2022.02.12.480214; PMID: 34435732). We further note that some trajectory programs require ‘supervision’ (e.g., specifying a root and/or roots in Monocle), and many do not allow for consideration and removal of contaminating ambient RNA, whose elimination is an important part of our analytic pipeline.

Our work focuses on determining the “ground truth”, which we validated by independent, multidimensional approaches including scRNA-seq, chemical and genetic lineage tracing, and high-resolution 3D imaging of protein and mRNA. Together, these data unequivocally establish that brown adipocyte neogenesis occurs from the proliferation and differentiation of PDGFRA progenitors, whose trajectory involves activation of quiescent ASC1 cells, proliferation, and differentiation into mature brown adipocytes. We are therefore reluctant to select an algorithm that supports our experimental data post hoc because believe such an approach would be inherently biased. Rather, we believe our data will provide a valuable resource for computational biologists interested in adipose tissue biology to validate assumptions and further benchmark their methods to our dataset.

Reviewer #2 (Recommendations for the authors):1. Please explain why mice were adapted to RT rather than thermoneutrality for the control mice used in the single cell experiments (since RT mice are also cold)? Could this affect interpretations?

We have performed preliminary scRNA-seq experiments (not shown) at thermoneutral conditions (30°C) compared to RT and cold treatment in the context of MACs-purified PDGFRA cells. Neogenesis is virtually absent at both 30 ^o^C and RT and thus differences are not informative with respect to neogenesis induction. Comparison of cells under these two ‘basal’ conditions indicate they are highly similar, with the major difference seen in the expression of genes involving cholesterol biosynthesis (higher at 30 ^o^C and further downregulated at RT or by cold).

2. It would be nice to include functional data showing that the ASC1 population is the exclusive source of brown adipocytes.

We agree that specific genetic tracing of ASC1 cells would further support our conclusion that these cells are the major, if not exclusive, source of cold-induced brown adipocytes (BAs); however, such tracing would require intersectional genetics that are not yet available. We are certainly open to the possibility that BAs can be derived from other sources under other conditions (e.g., development). Our current and past results establish that essentially all cold-induced BAs in adult mice are derived from stomal cells with an immediate history of PDGFRA expression. Of the cells that express PDGFRA, scRNA-seq established that only ASC1 give rise to cells in the brown adipogenic trajectory. Using smFISH, we further identified adipogenic niches of proliferating and differentiating cells that co-express the ASC1-specific marker Bmper, lending further experimental support for ASC1 as an immediate progenitor.

3. It was surprising that the authors did not test if KO of Adrb1 in other populations (e.g. BAs) inhibits NE-stimulated neogenesis as they showed by global KO that it is required. What's the key cell population?

We believe this is an important question that will require further experimentation to provide conclusive results. As discussed, the simplest interpretation based upon available data is that activation of either ADRB1 (cold) or ADRB3 (CL treatment) in BAs is sufficient to induce immune cell recruitment and proliferation and differentiation, although the extent to which this occurs is slightly variable among animals and the type of stimuli administered. We acknowledge that we have not demonstrated the necessity of ADRB1 in BAs but believe this to be an important concept for a follow-up study.

4. These data are beautiful and establish interesting hypotheses, though I would have liked to see more functional analysis of potential mechanisms, for example between immune-BA interactions. Nevertheless, I do think this is appropriate for eLife.

We agree that the present work provides a foundation and framework for addressing immune cell subtypes and cytokine signaling mechanisms. We are actively working in this area and hope to have results to report in the near future. Although we are working in this domain, these efforts extend beyond the scope of the present work.